# Multi-view Latent Diffusion Reconstruction for Vision-enhanced Time Series Forecasting

## Abstract

Recent studies have explored diffusion models for time series forecasting, yet most methods operate directly on 1D signals and tend to overlook intrinsic temporal structures (e.g., periodicity and trend). This often leads to suboptimal long-range dependency modeling and poorly calibrated uncertainty. To this end, we propose LDM4TS, a vision-enhanced time series forecasting framework that visualizes time series into structured 2D representations and leverages the image reconstruction capabilities of diffusion models. Raw sequences are first converted into complementary visual inputs, forming multiple views that collectively capture diverse temporal structures. By leveraging the generative nature of the diffusion process, the framework not only yields accurate point forecasts but also provides the capability to characterize predictive uncertainty. Extensive experiments demonstrate that LDM4TS outperforms various specialized forecasting models for time series forecasting tasks. The source code is available at: https://anonymous.4open.science/r/LDM4TS-53FB/.

## 1 Introduction

Time Series Forecasting (TSF) is a critical capability across many real-world domains Jin et al. (2024a), enabling proactive decisions in demand planning Leonard (2001), energy load scheduling Liu et al. (2023), climate and environmental modeling Schneider & Dickinson (1974), and traffic flow management Zheng et al. (2006). As temporal data grow in scale and heterogeneity, practitioners increasingly require models that are robust across regimes.

Deep learning has substantially advanced TSF by learning complex temporal dependencies. Early recurrent models introduced sequential inductive biases Cho et al. (2014); Hochreiter & Schmidhuber (1997); Lin et al. (2024b), while Transformer-based architectures improved long-range modeling and computational efficiency Nie et al. (2023a); Zhou et al. (2021; 2022); Wu et al. (2021); Woo et al. (2022); Liu et al. (2024). More recently, leveraging pre-trained or foundation models has shown promise in time series Zhou et al. (2023); Jin et al. (2024b); Zhong et al. (2025). Despite these advances, these methods still operate on raw 1D inputs, struggle to capture intrinsic temporal structures and model uncertainty for stable long-horizon forecasting and robust generalization.

To address this, diffusion models have been introduced as powerful generative frameworks for structure-aware reconstruction and uncertainty modeling in TSF Rasul et al. (2021a); Shen et al. (2024); Shen & Kwok (2023); Tashiro et al. (2021); Yan et al. (2021). Denoising Diffusion Probabilistic Models (DDPMs) progressively remove noise and sample diverse yet realistic images Ho et al. (2020), while Latent Diffusion Models (LDMs) partially alleviate quadratic computational cost by operating in a compressed latent space Rombach et al. (2022). Though their great success in vision tasks like image-to-image Saharia et al. (2022a;b); Meng et al. (2021); Mokady et al. (2023); Zhang et al. (2023), these models struggle to capture the sequential nature of time series and often fail to preserve long-range temporal dependencies when directly applied to raw 1D signals.

To harness diffusion models without sacrificing temporal structure, an intuitive idea is to transform sequences into compact 2D visual representations that encode local trends, periodicity, and cross-channel interactions as diverse spatial textures Chen et al. (2024). Early studies have also shown that time series data can be transformed into coherent visual representations, although most adopt a single-view perspective that preserves only specific temporal characteristics Eckmann et al. (1995); van den Oord et al. (2016); Wang & Oates (2015a); Griffin & Lim (1984); Daubechies (2002); Vetterli

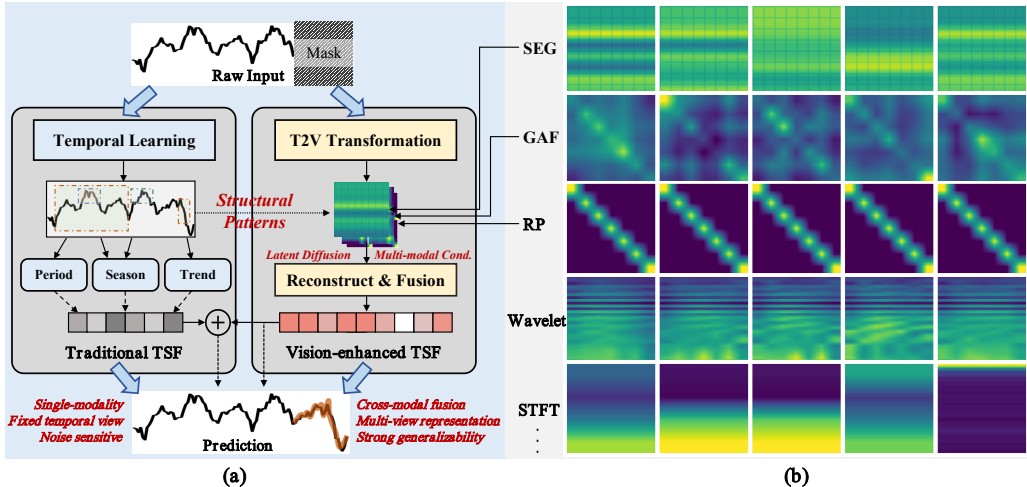

Figure 1: (a) Comparison between traditional TSF methods and vision-enhanced approach. (b) Illustration of different transformations for time series data, each with unique temporal patterns.

& Herley (1992). These vision-based transformations introduce spatial inductive biases, allowing diffusion models to fully exploit their strengths in uncertainty modeling while capturing both local and global temporal dependencies essential for robust time series forecasting Wu et al. (2022); Chen et al. (2024); Zhong et al. (2025). However, several critical limitations exist: (i) Most existing methods based on single-view transformation leave inherent structural patterns under-represented, necessitating a multi-view strategy to extract complementary structural patterns; (ii) The integration of accurate point forecasting and quantification remains largely unexplored in vision-based approaches.

To address these challenges, we present LDM4TS, the first attempt to leverage multi-view transformation and latent diffusion reconstruction as a vision-enhanced time series forecasting method. Our approach proposes the image multi-modal conditional reconstruction to enhance the TSF task, as illustrated in Figure 1 (a). LDM4TS combines the strong reconstruction capability of diffusion models with multi-view vision-enhanced temporal dependency learning. Specifically, ❶ we first transform raw time series data into multi-view visual representations, including multiple Time-to-Vision (T2V) transformation strategies to capture a full spectrum of temporal structures. ❷ These visual representations are then mapped into a low-dimensional latent space, where a latent diffusion model progressively denoises the latent variables. ❸ To further increase the flexibility of the model, the diffusion process is conditioned on the frequency embedding and textual embedding to capture domain-specific knowledge or statistical properties of the time series via cross-attention. ❹ Finally, a projection and fusion module is introduced to extract complex dependencies from the reconstructed representations and predict future time series. The key contributions of this work are as follows:

**1) Multi-view Visual Representations:** We present the first work that transforms time series into multi-view visual representations with preserved crucial temporal properties, thus leveraging diffusion models' power to capture the complex temporal structures and intrinsic patterns.

**2) Vision-enhanced Latent Diffusion Framework for TSF:** We develop LDM4TS, a unified framework that reconstructs multi-view T2V representations via latent diffusion and a multi-modal conditional-guided mechanism for effective time series forecasting.

**3) Comprehensive Empirical Validation:** Extensive experiments verify that LDM4TS achieves state-of-the-art performance on diverse datasets, outperforming specialized TSF models and methods with pre-trained components on time series forecasting tasks.

## 2  RELATED WORK

**Diffusion Models for Time Series.**  Diffusion models have emerged as a powerful class of generative approaches, demonstrating remarkable success across various high-dimensional data domains.

Denoising Diffusion Probabilistic Models (DDPMs) Ho et al. (2020) employ a Markov chain to add and remove Gaussian noise, progressively generating high-fidelity samples. Score-based diffusion models Song et al. (2020) directly estimate the score function of data distributions, while conditional diffusion models Dhariwal & Nichol (2021) further incorporate guidance signals to steer the generative process. Recent years have witnessed increasing applications of diffusion models in time series analysis Yang et al. (2024b); Lin et al. (2024a). TimeGrad Rasul et al. (2021a) pioneered the integration of diffusion with autoregressive modeling, and D3VAE Li et al. (2022) combines variational autoencoders with diffusion for improved flexibility. TSDiff Kollovieh et al. (2024) iteratively refines probabilistic forecasts through implicit densities. In the context of imputation, conditional models such as CSDI Tashiro et al. (2021) and MIDM Wang et al. (2023a) leverage conditional score matching for irregular time series. TimeDiff Shen & Kwok (2023) introduces a non-autoregressive conditional diffusion framework for time series prediction, while TMDM Li et al. (2024) employs transformer-modulated diffusion for multivariate probabilistic forecasting. DiffusionTS Yuan & Qiao (2024) focuses on interpretable diffusion for general time series generation, and NsDiff Ye et al. (2025) specifically addresses non-stationary time series forecasting via specialized diffusion modeling techniques. Domain-specific designs have also emerged, such as DiffLoad Wang et al. (2023b) for load forecasting, WaveGrad Chen et al. (2020) and DiffWave Kong et al. (2020) for audio synthesis, and EHRDiff Yuan et al. (2023) for healthcare applications. DiffSTG Wen et al. (2023) further explores spatio-temporal graph structures in diffusion models for time series.

*However, most existing diffusion methods focus on single-modality or lack mechanisms for leveraging multi-view visual representations for TSF.* Our work advances the development of latent diffusion models for TSF by incorporating multi-modal information and exploiting cross-modal conditioning mechanisms, thereby substantially improving the accuracy and robustness under different scenarios.

**Vision-enhanced Time Series Forecasting.** Vision models like ViT Dosovitskiy (2020) and MAE He et al. (2022) have demonstrated remarkable success in computer vision through their exceptional feature extraction capabilities, demonstrating unprecedented generalization power when pre-trained on large-scale datasets like ImageNet Deng et al. (2009). Inspired by the success of these vision models, researchers have begun exploring their potential in time series forecasting.

Leveraging vision models for time series analysis has recently gained increasing traction Ni et al. (2025); Zhao et al. (2025). Early methods of treating time series as images have evolved from traditional approaches using Gramian Angular Fields (GAF), Markov Transition Fields (MTF) Wang & Oates (2015b) or various spectrogram-based approaches van den Oord et al. (2016); Griffin & Lim (1984); Daubechies (2002). These methods enable the use of 2D vision models like CNNs Wang & Oates (2015b); Barra et al. (2020), and more sophisticated transformer-based methods Dosovitskiy (2020); Wu et al. (2022); Chen et al. (2024) for time series tasks. TimesNet Wu et al. (2022) exploits 2-D matrix representations, and VisionTS Chen et al. (2024) utilizes pre-trained visual architectures for effective feature extraction and transfer learning. ViTime Yang et al. (2024a) demonstrates the possibility of zero-shot forecasting by treating time series as visual signals.

*However, these approaches are predominantly deterministic and lack uncertainty quantification capabilities since they are not built within generative frameworks.* Our work addresses these limitations by integrating latent diffusion models with visual representations in a unified framework. This design enables our model to effectively capture temporal dependencies while maintaining the uncertainty modeling capabilities inherent in diffusion models.

## 3 METHODOLOGY

As illustrated in Figure 2, LDM4TS employs a cross-modal architecture that leverages the visual pattern reconstruction capabilities of latent diffusion models to enhance time series forecasting. The framework operates through three key stages: ❶ First, it transforms raw time series data into multi-view visual representations using complementary encoding methods, each capturing underlying characteristics. ❷ Second, these visual representations are processed through a latent diffusion model that iteratively denoises the multi-modal encoded data, guided by frequency and textual conditions that provide domain knowledge and statistical context. ❸ Finally, the model combines the diffusion-generated features with explicit temporal features through an adaptive fusion mechanism, producing accurate and robust forecasts that capture both global patterns and local dynamics.

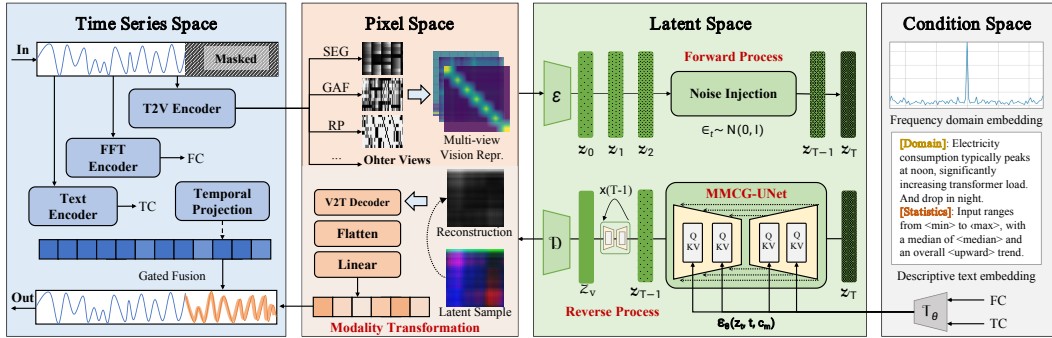

Figure 2: The framework of our proposed LDM4TS. Time series data is first transformed into complementary visual representations (SEG: Segmentation, GAF: Gramian Angular Field, RP: Recurrence Plot) that encode structural temporal patterns. A conditional latent diffusion model then reconstructs the images through iterative denoising guided by a multi-modal conditional-guided mechanism (FC: frequency conditioning, TC: textual conditioning). Finally, the reconstructed images are mapped back to time series space with explicit temporal dependencies and implicit patterns.

## 3.1 MULTI-VIEW VISION TRANSFORMATION FOR TIME SERIES

Time series data exhibits complex temporal patterns across multiple views, from local fluctuations to long-term trends, making structural modeling challenging. We propose a novel approach that transforms time series into multiple complementary visual representations, each capturing unique temporal characteristics through a T2V vision encoder (VE) integrated with multiple transformation strategies and multi-scale convolution combination. Given an input sequence $X \in \mathbb{R}^{B \times L \times D}$, where $B$ denotes the batch size, $L$ represents the sequence length, and $D$ indicates the feature dimension, we construct a multi-channel image representation $I_\phi$ through complementary encoding methods $\phi(\cdot)$. The technical details of all the transformation processes are presented in Appendix E.

We implemented 8 different transformation strategies and simplified their utilization. For illustrative and evaluation purposes, we generate three visual channels with strong complementary properties to validate our approach. Specifically, One combination of the T2V encoder transformation process is inspired by (i) the Segmentation representation (SEG) Chen et al. (2024) that employs periodic restructuring to preserve local temporal structures, enabling the detection of recurring patterns across multiple time scales; (ii) the Gramian Angular Field (GAF) Zheng et al. (2014); Wang & Oates (2015a) that transforms temporal correlations into spatial patterns through polar coordinate mapping, effectively capturing long-range dependencies crucial for forecasting; and (iii) the Recurrence Plot (RP) Eckmann et al. (1995); Marwan et al. (2007) that constructs similarity matrices between time points to reveal both cyclical behaviors and temporal anomalies, providing a complementary view of the underlying structure. As demonstrated in Figure 1 (b), these three visual encoding strategies effectively convert temporal dynamics into structured spatial patterns, enabling our model to capture history dependencies and underlying features. The transformation process is formulated as follows:

$$\tilde{X} = \frac{X - \min(X)}{\max(X) - \min(X) + \epsilon}, \quad I'_\phi = \mathcal{F}(\mathcal{R}(\mathcal{P}(\tilde{X}^T), \lceil \frac{L+p}{P} \rceil, P)), \tag{1}$$

$$I''_\phi = \mathcal{F}(\frac{1}{D}\sum_{d=1}^{D}\cos(\theta_d \oplus \theta_d^T)), \quad I'''_\phi = \mathcal{F}(\exp(-\frac{\|X_i - X_j\|_2^2}{2\sigma^2})), \tag{2}$$

$$I_\phi = \text{VE}(X, \phi) = \text{Multi-Conv}(\text{Concat}[I'_\phi; I''_\phi; I'''_\phi]) \in \mathbb{R}^{B \times 3 \times H \times W}, \tag{3}$$

where $\epsilon = 1e^{-8}$ is a small constant added to prevent division by zero during normalization; $\mathcal{P}(\cdot)$ represents padding operation that ensures the sequence length is divisible by periodicity $P$ and $p$ is the padding length; $\mathcal{R}(\cdot)$ restructures the padded sequence into a 2D matrix with dimensions determined by $T$; $\mathcal{F}(\cdot)$ performs bilinear interpolation to the target size $(H, W)$ and normalizes to $[0, 1]$; $\theta_d = \arccos(2\tilde{X}_{:,:,d} - 1)$ represents the angular coordinates of the normalized time series mapped to $[-1, 1]$; $\oplus$ denotes the outer sum operation generating pairwise temporal correlations;

$X_i$ and $X_j$ refer to phase space vectors at time points $i$ and $j$ respectively. and $\sigma$ is the standard deviation of these distances. The final multi-channel image $I_\phi$ integrates three complementary views of temporal dynamics with standard image shape $3 \times H \times W$.

## 3.2 Multi-modal Conditional-guided Latent Diffusion for Reconstruction

Traditional diffusion models operate in high-dimensional pixel space, making them computationally intensive for time series. We complete the masked future region of the structured 2D encodings in a low-dimensional space via a lightweight latent diffusion model. Beyond standard latent diffusion Rombach et al. (2022), our denoiser is specifically built Multi-modal Conditional-Guided U-Net (**MMCG-UNet**) that projects heterogeneous conditions (frequency/text) into a shared guidance space with fusion and normalization to preserve global structures. This design yields stronger structure completion and better calibration at a few sampling steps. More algorithm details are in Appendix D.

**Multi-conditional Generation Framework.** To guide accurate temporal feature reconstruction, we implement a cross-modal conditioning mechanism that integrates both frequency domain information and semantic descriptions. Given a visual representation $I \in \mathbb{R}^{B \times 3 \times H \times W}$, we first encode it into latent space and derive conditional signals as:

$$c_{freq} = \text{FFTEncoder}(X), \quad c_{text} = \text{TextEncoder}(X), \tag{4}$$

$$z_0 = \mathcal{E}(I_\phi) \cdot \mathbf{s}, \quad c_m^{(t)} = \text{CrossAttn}(\text{MLP}([c_{text}; c_{freq}]), z_t), \tag{5}$$

where $\mathcal{E}(\cdot)$ represents the frozen pre-trained VAE, $\mathbf{s}$ is the latent space scaling factor (see Appendix D.1 for detailed derivation). $c_{freq} \in \mathbb{R}^{B \times (2DL+2)}$ captures periodic patterns through frequency analysis while $c_{text} \in \mathbb{R}^{B \times d_{model}}$ encodes statistical properties and domain knowledge through natural language descriptions. For inference step $t$, condition $c_m$ is updated with denoised $z_t$. Our framework provides flexibility for integrating multi-modal conditional embeddings across. The detailed implementations of the aforementioned FFTEncoder and TextEncoder are provided in Appendix D.4.

**Forward Diffusion Process.** Our forward process implements a variance-preserving Markov chain that gradually injects Gaussian noise into the latent representations. By operating in compressed latent space rather than pixel space, this approach enables efficient learning of temporal patterns across different scales while preserving the intrinsic information from vision transformations. For a given initial latent representation $z_0$, we define the forward diffusion process distribution $q$ as:

$$q(z_t | z_{t-1}, I_\phi) = \mathcal{N}(z_t; \sqrt{\alpha_t} z_{t-1}, (1 - \alpha_t)\mathbf{I}), \tag{6}$$

$$q(z_t | z_0, I_\phi) = \mathcal{N}(z_t; \sqrt{\bar{\alpha}_t} z_t, (1 - \bar{\alpha}_t)\mathbf{I}), \quad \bar{\alpha}_t = \prod_{s=1}^{t} \alpha_s, \quad t \in \{1, ..., T\}, \tag{7}$$

where $\{\alpha_t\}_{t=1}^{T}$ defines a scaled linear noise schedule, and $\bar{\alpha}_t$ controls the cumulative noise level across $t$ timesteps. $\mathcal{N}$ denotes a multivariate Gaussian distribution.

**MMCG-UNet De-noising Process.** The reverse process employs a parameterized U-Net architecture to denoise the representations, exploiting cross-modal conditioning mechanisms. By incorporating frequency and semantic embeddings, this process uniquely captures complex temporal dynamics while maintaining coherent long-term dependencies. The denoising process is formulated as:

$$p_\theta(z_{t-1} \mid z_t, c_m^{(t)}) = \mathcal{N}(z_{t-1}; \mu_\theta(z_t, t, c_m^{(t)}), \Sigma_\theta(z_t, t)), \tag{8}$$

$$\mu_\theta(z_t, t, c_m^{(t)}) = \frac{1}{\sqrt{\alpha_t}} \left( z_t - \frac{1 - \alpha_t}{\sqrt{1 - \bar{\alpha}_t}} \epsilon_\theta(z_t, t, c_m^{(t)}) \right), \tag{9}$$

where $\epsilon_\theta$ is the MMCG-UNet that predicts the noise sample given the noisy latent $z_t$, timestep $t$, and cross-modal condition embedding $c_m$. We pre-compute and cache diffusion parameters including $\alpha_t$, $\sqrt{\alpha_t}$, cumulative products $\bar{\alpha}_t$ to improves training and inference efficiency. The reconstructed image $\hat{I}_t = \mathcal{D}(z_t/s)$ is obtained by decoding the denoised latent representation through the VAE decoder $\mathcal{D}(\cdot)$, and the visual feature $z_v = \text{MMCG}(\hat{I}_0)$ is computed via up/downsampling followed by GeLU.

### 3.3 MULTI-MODAL PREDICTION AND OPTIMIZATION

**Multi-modal Feature Fusion.** While the latent diffusion model captures global patterns effectively, local temporal dynamics and distribution shifts require explicit modeling. As shown in Fig. 3, we utilize a temporal projection (TP) that complements the diffusion process through three key components: patch embedding, attention-based projection, and multi-modal gated fusion. Given input sequence $X \in \mathbb{R}^{B \times L \times D}$, we adopt the patch embedding strategy Dosovitskiy (2020); Nie et al. (2023b) to encode temporal hidden states, which are then processed through $l$ layers encoders, where $X_{norm} = \text{LN}(X)$. The resulting embeddings are constructed as follows:

$$h_0 = \text{Embed}(X_{norm}) \in \mathbb{R}^{B \times N_p \times d}, \quad h_l' = h_{l-1} + \text{MSA}(\text{LN}(h_{l-1})), \quad (10)$$

$$h_l = h_l' + \text{MLP}(\text{LN}(h_l')), \quad z_h = \text{Linear}(h_l) \in \mathbb{R}^{B \times L_{pred} \times D}, \quad (11)$$

where $N_p$ denotes patch count, $h$ is hidden states and $d$ is the hidden dimension. $\text{MSA}(\cdot)$ and $\text{LN}(\cdot)$ represent multi-head self-attention and layer normalization respectively.

**Forecasting and Optimization Objective.** We employ a gated fusion mechanism to combine temporal features $z_h$ and visual features $z_v$ derived from the reconstruction $\hat{I}$ for point predictions:

$$z_v = \text{Linear}(\text{Flatten}(\hat{\mathcal{F}}(\hat{I}))), \quad (12)$$

$$g = \sigma(\text{MLP}([z_h; z_v])), \quad (13)$$

$$\hat{Y} = g \odot z_h + (1 - g) \odot z_v, \quad (14)$$

where $\sigma$ denotes the activation function, $\odot$ represents element-wise multiplication, and $g$ are learnable gating weights that dynamically balance the contributions from each modality. The operator $\hat{\mathcal{F}}(\cdot)$ denotes the inverse of $\mathcal{F}(\cdot)$, a V2T decoder that maps visual representations back to the original value range of the normalized time series.

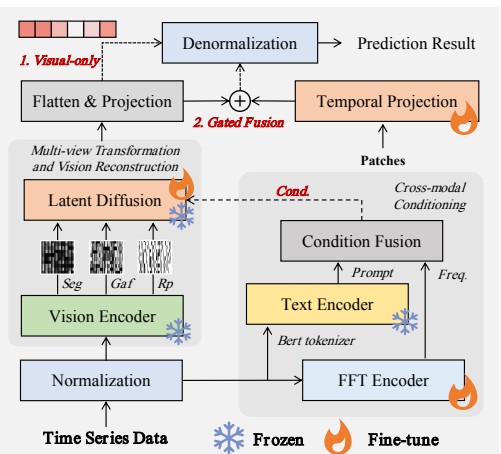

Figure 3: The forward process of LDM4TS.

The model is trained using mean squared error (MSE) loss for point prediction. For probabilistic forecasting in Appendix C, we exploit the inherent stochasticity of the diffusion process to generate a predictions set $\{\hat{Y}^{(s)}\}_{s=1}^{S}$ by sampling $S$ latent trajectories. Our implementation leverages the deterministic nature of time series forecasting while accounting for inherent uncertainties, providing accurate predictions and well-calibrated prediction intervals.

## 4 EXPERIMENTS

### 4.1 SETTINGS

**Dataset and Metrics.** We evaluate LDM4TS on seven widely used time series datasets spanning diverse domains, including energy consumption (ETTh1, ETTh2, ETTm1, ETTm2), weather forecasting, and electricity load prediction (ECL; 321 variables), Zhou et al. (2021); Lai et al. (2018). These benchmarks are widely adopted for long-term forecastingWu et al. (2022) and cover a range of sampling frequencies, dimensionalities, and temporal structures. These datasets are chosen for their varying characteristics in terms of sampling frequency, dimensionality, and temporal patterns. Our experiments primarily focus on point forecasting, evaluated by Mean Absolute Error (MAE) and Mean Squared Error (MSE), following standard practice. Due to space constraints, we additionally reported performance on irregular time series in Appendix C.1 and evaluated the Quantile Interval Calibration Error (QICE) Han et al. (2022) for probabilistic forecasting against diffusion-based models and report comparative results in Appendix C. Further dataset and metric details are provided in Appendices A.1 and A.3.

Table 1: Long-term forecasting results. All results are averaged from four forecasting horizons: $H \in \{96, 192, 336, 720\}$. A lower value indicates better performance. **Red**: best, Blue: second best.

| Methods | LDM4TS (Ours) | | ETSformer (2022) | | Stationary (2022b) | | Autoformer (2021) | | FEDformer (2022) | | DLinear (2023) | | Informer (2021) | | TimesNet (2022) | | LightTS (2023) | | Reformer (2020) | | PatchTST (2023b) | | GPT4TS (2023) | |
|---|---|---|---|---|---|---|---|---|---|---|---|---|---|---|---|---|---|---|---|---|---|---|---|---|
| Metric | MSE | MAE | MSE | MAE | MSE | MAE | MSE | MAE | MSE | MAE | MSE | MAE | MSE | MAE | MSE | MAE | MSE | MAE | MSE | MAE | MSE | MAE | MSE | MAE |
| ETTh1 | **0.439** | **0.452** | 0.542 | 0.510 | 0.570 | 0.537 | 0.504 | 0.492 | 0.440 | 0.460 | 0.460 | 0.457 | 1.040 | 0.799 | 0.460 | 0.455 | 0.590 | 0.544 | 1.006 | 0.745 | 0.468 | 0.454 | 0.465 | 0.455 |
| ETTh2 | **0.377** | **0.412** | 0.439 | 0.452 | 0.526 | 0.516 | 0.467 | 0.468 | 0.439 | 0.451 | 0.564 | 0.519 | 4.551 | 1.742 | 0.407 | 0.421 | 1.260 | 0.678 | 2.531 | 1.244 | 0.408 | 0.425 | 0.381 | 0.412 |
| ETTm1 | **0.349** | **0.385** | 0.429 | 0.425 | 0.481 | 0.456 | 0.576 | 0.526 | 0.471 | 0.470 | 0.404 | 0.408 | 0.867 | 0.690 | 0.477 | 0.443 | 0.427 | 0.437 | 1.013 | 0.737 | 0.387 | 0.401 | 0.388 | 0.403 |
| ETTm2 | **0.283** | **0.329** | 0.293 | 0.342 | 0.306 | 0.347 | 0.307 | 0.351 | 0.318 | 0.366 | 0.304 | 0.349 | 1.593 | 0.908 | 0.299 | 0.333 | 0.830 | 0.614 | 1.874 | 1.009 | 0.293 | 0.337 | 0.284 | 0.339 |
| Weather | **0.229** | **0.277** | 0.271 | 0.334 | 0.288 | 0.314 | 0.329 | 0.375 | 0.333 | 0.375 | 0.246 | 0.306 | 0.634 | 0.549 | 0.265 | 0.288 | 0.259 | 0.315 | 1.229 | 0.858 | 0.258 | 0.281 | 0.264 | 0.284 |
| ECL | **0.182** | **0.273** | 0.208 | 0.323 | 0.193 | 0.296 | 0.253 | 0.352 | 0.612 | 0.377 | 0.225 | 0.319 | 0.378 | 0.438 | 0.208 | 0.303 | 0.243 | 0.343 | 0.326 | 0.404 | 0.188 | 0.275 | 0.205 | 0.290 |

Table 2: Few-shot learning on 10% training data. We use the same protocol in Table 1.

| Methods | LDM4TS (Ours) | | ETSformer (2022) | | Stationary (2022b) | | Autoformer (2021) | | FEDformer (2022) | | DLinear (2023) | | Informer (2021) | | TimesNet (2022) | | LightTS (2023) | | Reformer (2020) | | iTransformer (2024) | | PatchTST (2023b) | |
|---|---|---|---|---|---|---|---|---|---|---|---|---|---|---|---|---|---|---|---|---|---|---|---|---|
| Metric | MSE | MAE | MSE | MAE | MSE | MAE | MSE | MAE | MSE | MAE | MSE | MAE | MSE | MAE | MSE | MAE | MSE | MAE | MSE | MAE | MSE | MAE | MSE | MAE |
| ETTh1 | **0.471** | **0.468** | 1.180 | 0.834 | 0.915 | 0.639 | 0.701 | 0.596 | 0.638 | 0.561 | 0.691 | 0.599 | 1.199 | 0.808 | 0.869 | 0.628 | 1.375 | 0.877 | 1.249 | 0.833 | 0.660 | 0.551 | 0.633 | 0.542 |
| ETTh2 | **0.371** | **0.405** | 0.894 | 0.713 | 0.462 | 0.455 | 0.488 | 0.499 | 0.466 | 0.475 | 0.605 | 0.538 | 3.871 | 1.513 | 0.479 | 0.465 | 2.655 | 1.159 | 3.485 | 1.486 | 0.435 | 0.439 | 0.415 | 0.431 |
| ETTm1 | **0.371** | **0.393** | 0.980 | 0.714 | 0.797 | 0.578 | 0.802 | 0.628 | 0.721 | 0.605 | 0.411 | 0.429 | 1.192 | 0.820 | 0.479 | 0.465 | 0.970 | 0.704 | 1.426 | 0.856 | 0.450 | 0.431 | 0.501 | 0.466 |
| ETTm2 | **0.270** | **0.331** | 0.447 | 0.487 | 0.332 | 0.366 | 1.341 | 0.930 | 0.463 | 0.488 | 0.316 | 0.368 | 3.369 | 1.439 | 0.319 | 0.353 | 0.987 | 0.755 | 3.978 | 1.587 | 0.305 | 0.349 | 0.296 | 0.343 |
| Weather | **0.229** | **0.276** | 0.318 | 0.360 | 0.318 | 0.323 | 0.300 | 0.342 | 0.284 | 0.283 | 0.241 | 0.283 | 0.597 | 0.494 | 0.279 | 0.301 | 0.289 | 0.322 | 0.526 | 0.469 | 0.272 | 0.290 | 0.242 | 0.279 |

Table 3: Few-shot learning on 5% training data. **Red**: best, Blue: second best.

| Methods | LDM4TS (Ours) | | ETSformer (2022) | | Stationary (2022b) | | Autoformer (2021) | | FEDformer (2022) | | DLinear (2023) | | Informer (2021) | | TimesNet (2022) | | LightTS (2023) | | Reformer (2020) | | PatchTST (2023b) | |
|---|---|---|---|---|---|---|---|---|---|---|---|---|---|---|---|---|---|---|---|---|---|---|
| Metric | MSE | MAE | MSE | MAE | MSE | MAE | MSE | MAE | MSE | MAE | MSE | MAE | MSE | MAE | MSE | MAE | MSE | MAE | MSE | MAE | MSE | MAE |
| ETTh1 | **0.458** | **0.456** | 0.398 | 0.850 | 0.662 | 1.026 | 0.722 | 0.599 | 0.659 | 0.562 | 0.750 | 0.611 | 1.225 | 0.817 | 0.926 | 0.648 | 1.451 | 0.903 | 1.242 | 0.835 | 0.687 | 0.722 |
| ETTh2 | **0.388** | **0.412** | 0.809 | 0.681 | 0.470 | 0.489 | 0.470 | 0.489 | 0.441 | 0.457 | 0.826 | 0.616 | 3.923 | 1.654 | 0.464 | 0.454 | 3.206 | 1.268 | 3.527 | 1.473 | 0.827 | 0.615 |
| ETTm1 | 0.407 | **0.412** | 1.125 | 0.783 | 0.857 | 0.598 | 0.796 | 0.621 | 0.731 | 0.593 | **0.401** | 0.417 | 1.163 | 0.791 | 0.717 | 0.561 | 1.123 | 0.766 | 1.264 | 0.827 | 0.526 | 0.476 |
| ETTm2 | **0.311** | **0.353** | 0.534 | 0.547 | 0.341 | 0.373 | 0.388 | 0.433 | 0.381 | 0.404 | 0.399 | 0.426 | 3.658 | 1.489 | 0.345 | 0.373 | 1.416 | 0.871 | 3.582 | 1.487 | 0.315 | 0.353 |
| Weather | **0.258** | **0.294** | 0.333 | 0.371 | 0.327 | 0.328 | 0.311 | 0.354 | 0.310 | 0.353 | 0.264 | 0.309 | 0.584 | 0.528 | 0.298 | 0.318 | 0.306 | 0.345 | 0.447 | 0.453 | 0.269 | 0.303 |

**Compared Methods.** We compared point forecasting with a set of recent competitive models, including ❶ time-series specific models PatchTST Nie et al. (2023b), FEDformer Zhou et al. (2022), Autoformer Wu et al. (2021), Informer Zhou et al. (2021), ETSformer Woo et al. (2022), Reformer Kitaev et al. (2020), DLinear Zeng et al. (2023), TimesNet Wu et al. (2022), ESTformer Woo et al. (2022), Non-Stationary Transformer Liu et al. (2022a), LightTS Zhang et al. (2022), and ❷ advanced models like PatchTST Nie et al. (2023b), iTransformer Liu et al. (2024), Timemixer++ Wang et al. (2024a), FITS Xu et al. (2023) and TimeVLM Zhong et al. (2025), VisionTS Chen et al. (2024), GPT4TS Zhou et al. (2023) with pre-trained components. ❸ For probabilistic forecasting, we selected six strong baselines including TimeGrad Rasul et al. (2021a), CSDI Tashiro et al. (2021), TimeDiff Shen & Kwok (2023), TMDM Li et al. (2024), DiffusionTS Yuan & Qiao (2024) and NsDiff Ye et al. (2025). More details of these methods are in Appendix B.

**Implementation Details.** The models are trained using the Adam optimizer with a learning rate of $10^{-3}$, batch size of 32, and a maximum of 10 epochs, applying an early stopping strategy. All experiments are conducted on an Nvidia RTX A6000 GPU with 48GB memory. All training and model parameter settings are detailed in Appendix A.2.

## 4.2 RESULTS

**Long-term Forecasting.** We evaluate the long-term forecasting capabilities of LDM4TS across multiple prediction horizons. As shown in Table 1, LDM4TS consistently outperforms state-of-the-art baselines. On the ETT datasets family, our approach demonstrates significant improvements, achieving the best MSE of 0.349 on ETTm1 compared to the second-best performer GPT4TS (0.381), and reducing MSE by 7.37% on ETTh2 (0.377) compared to TimesNet (0.407). The advantages extend to high-dimensional scenarios, achieving superior results on Electricity (321 variables, MSE: 0.182 vs PatchTST 0.188). Overall, LDM4TS achieves competitive performances among these datasets, validating that our vision-enhanced modeling strategy effectively captures complex temporal dynamics across diverse forecasting scenarios.

**Few-shot Forecasting.** To evaluate model robustness under data scarcity, we conduct experiments using only 10% and 5% of the training data. As shown in Table 2, LDM4TS achieves optimal or the

Table 4: Zero-shot learning results among the ETT dataset family. **Red**: best, Blue: second best.

| Methods | LDM4TS (Ours) | | ETSformer (2022) | | Stationary (2022b) | | Autoformer (2021) | | FEDformer (2022) | | DLinear (2023) | | Informer (2021) | | ETSformer (2022) | | LightTS (2023) | | Reformer (2020) | | CSDI (2021) | |
|---|---|---|---|---|---|---|---|---|---|---|---|---|---|---|---|---|---|---|---|---|---|---|
| Metric | MSE | MAE | MSE | MAE | MSE | MAE | MSE | MAE | MSE | MAE | MSE | MAE | MSE | MAE | MSE | MAE | MSE | MAE | MSE | MAE | MSE | MAE |
| ETTh1→ETTh2 | **0.458** | **0.452** | 0.589 | 0.589 | 0.591 | 0.530 | 0.582 | 0.548 | 0.495 | 0.501 | 0.493 | 0.488 | 2.292 | 1.169 | 0.589 | 0.589 | 1.075 | 0.699 | 2.119 | 1.125 | 0.500 | 0.527 |
| ETTh1→ETTm2 | **0.369** | **0.400** | 0.569 | 0.568 | 0.437 | 0.439 | 0.457 | 0.483 | 0.373 | 0.424 | 0.415 | 0.452 | 2.167 | 1.124 | 0.569 | 0.568 | 1.058 | 0.700 | 2.228 | 1.165 | 0.410 | 0.444 |
| ETTm1→ETTh2 | **0.452** | **0.434** | 0.704 | 0.620 | 0.921 | 0.676 | 0.470 | 0.479 | 0.587 | 0.565 | 0.464 | 0.475 | 1.526 | 0.945 | 0.704 | 0.620 | 0.572 | 0.556 | 1.663 | 1.081 | 0.504 | 0.515 |
| ETTm1→ETTm2 | 0.354 | **0.367** | 0.603 | 0.578 | 0.493 | 0.470 | 0.469 | 0.484 | 0.424 | 0.463 | **0.335** | 0.389 | 1.521 | 0.951 | 0.603 | 0.578 | 0.466 | 0.495 | 2.017 | 1.111 | 0.405 | 0.440 |
| ETTm2→ETTh2 | 0.426 | 0.435 | 1.693 | 0.958 | 0.903 | 0.629 | **0.423** | 0.439 | 0.545 | 0.516 | 0.455 | 0.471 | 1.663 | 0.955 | 1.693 | 0.958 | 1.051 | 0.730 | 2.056 | 1.043 | 0.482 | 0.498 |
| ETTm2→ETTm1 | **0.588** | **0.487** | 0.728 | 0.607 | 1.055 | 0.796 | 0.755 | 0.591 | 0.819 | 0.618 | 0.649 | 0.537 | 0.854 | 0.637 | 0.728 | 0.607 | 0.716 | 0.550 | 0.941 | 0.698 | 1.039 | 0.763 |

Table 5: Ablation study results on different components on the Weather dataset. We compare the full LDM4TS model with variants excluding key components: latent diffusion model (w/o LDM), vision encoder (w/o VE), temporal encoder (w/o TE), textual conditioning (w/o TC), and frequency conditioning (w/o FC). We also investigate the impact of individual visual transformation methods. *%Deg* denotes the degradation percentage.

| Horizon | LDM4TS - Full | | w/o LDM | | w/o VE | | w/o TE | | w/o TC | | w/o FC | | w/o SEG | | w/o GAF | | w/o RP | |
|---|---|---|---|---|---|---|---|---|---|---|---|---|---|---|---|---|---|---|
| | MSE | MAE | MSE | MAE | MSE | MAE | MSE | MAE | MSE | MAE | MSE | MAE | MSE | MAE | MSE | MAE | MSE | MAE |
| 96 | 0.154 | 0.210 | 0.164 | 0.216 | 0.213 | 0.266 | 0.213 | 0.266 | 0.161 | 0.215 | 0.163 | 0.214 | 0.162 | 0.213 | 0.160 | 0.215 | 0.162 | 0.213 |
| 192 | 0.199 | 0.251 | 0.224 | 0.274 | 0.259 | 0.298 | 0.259 | 0.299 | 0.206 | 0.256 | 0.207 | 0.257 | 0.211 | 0.259 | 0.211 | 0.259 | 0.211 | 0.259 |
| 336 | 0.245 | 0.294 | 0.280 | 0.311 | 0.267 | 0.302 | 0.276 | 0.311 | 0.260 | 0.296 | 0.260 | 0.295 | 0.265 | 0.300 | 0.265 | 0.300 | 0.269 | 0.304 |
| 720 | 0.318 | 0.353 | 0.364 | 0.364 | 0.342 | 0.357 | 0.337 | 0.354 | 0.336 | 0.348 | 0.354 | 0.370 | 0.331 | 0.348 | 0.330 | 0.345 | 0.344 | 0.360 |
| Avg | 0.229 | 0.277 | 0.258 | 0.291 | 0.270 | 0.306 | 0.271 | 0.307 | 0.241 | 0.279 | 0.246 | 0.284 | 0.242 | 0.280 | 0.242 | 0.280 | 0.247 | 0.284 |
| %Deg | – | – | 12.66%↑ | 5.05%↑ | 18.00%↑ | 10.35%↑ | 18.42%↑ | 10.91%↑ | 5.05%↑ | 0.54%↑ | 7.47%↑ | 2.54%↑ | 5.78%↑ | 1.05%↑ | 5.47%↑ | 0.92%↑ | 7.60%↑ | 2.43%↑ |

second-best performance on all 5 datasets in both MSE and MAE metrics. LDM4TS outperforms the time series specific methods, with notable MSE reductions: 25.5% on ETTh1 (0.471 vs 0.630), 3.2% on ETTh2, and 9.7% on ETTm1 (0.371 vs 0.411). On the Weather dataset, LDM4TS outperforms the advanced methods like FEDformer and DLinear. Even with further reduced 5% training data, LDM4TS maintains strong performance by achieving the best results on 4 MSE and 5 MAE metrics across datasets. The robust performance under extreme data scarcity demonstrates how our vision-enhanced approach captures intrinsic patterns to address missing and sparse data challenges in real-world forecasting applications.

**Zero-shot Forecasting.** To evaluate cross-domain generalization, we conduct zero-shot transfer experiments across different datasets without any fine-tuning. As shown in Table 4, LDM4TS achieves the best performance in 4 MSE and 5 MAE metrics out of 6 scenarios, demonstrating strong cross-domain transferability. For challenging transfer tasks like $ETTh1 \rightarrow ETTh2$ and $ETTh1 \rightarrow ETTm2$, LDM4TS achieves MSE of 0.458 and 0.369 respectively, outperforming both DLinear (0.493, 0.415) and FEDformer (0.495, 0.373). The model also achieves the best on $ETTm1 \rightarrow ETTh1$ (0.452, 0.434) and $ETTm2 \rightarrow ETTm1$ (0.588, 0.487). The advantages are particularly pronounced when compared to other diffusion models like CSDI, with LDM4TS achieving from 9.9% to 36.1% improvements across all transfer scenarios. Notably, while most baselines exhibit significant performance degradation under cross-dataset transfer, LDM4TS maintains stable and competitive accuracy, underscoring its robust and reliable generalization capacity.

## 4.3 MODEL ANALYSIS

**Overall Performance Analysis.** LDM4TS demonstrates superior performance across various forecasting scenarios, excelling in long-term, few-shot, and zero-shot predictions, while maintaining computational efficiency with only 5.4M learnable parameters and fast inference speed (see Appendix F for detailed analysis). Through comprehensive experiments, we observe that our approach effectively captures both global trends and local patterns in time series data. As shown in Figure 4, LDM4TS achieves good performance in forecasting structured patterns, such as the clear periods in the Traffic datasets (MSE: 0.496) and regular consumption patterns in ECL data (MSE: 0.182). The performance shows slight degradation on datasets with irregular patterns or abrupt changes, suggesting potential areas for future improvement in handling non-stationary patterns.

**Visual Encoding Effectiveness.** For consistency across all experiments, we exclusively use a three-channel image representation composed of SEG, GAF, and RP transformations as our visual encoding strategy. The complementary nature of these encodings is particularly evident, and the combination achieves MSE reduction ranging from 5.6% to 7.5% compared to using any single

encoding method in the ETT datasets. Our framework provides a highly flexible architecture for combining different transformation strategies to extract intrinsic temporal features and preserve them within image structures. The current implementation supports various transformation methods beyond the three used in experiments, with detailed specifications and guidelines provided in Appendix E.

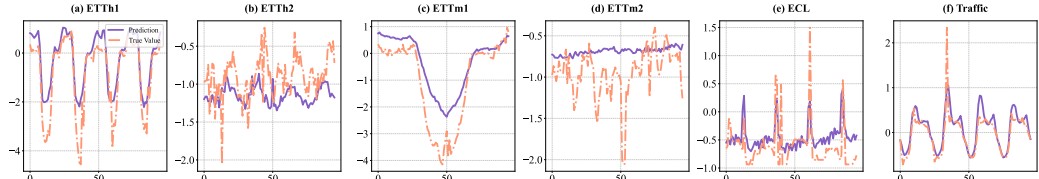

Figure 4: Qualitative visualization of long-term forecasting results generated by the proposed LDM4TS model across all benchmark datasets under the input-96-predict-96 setting.

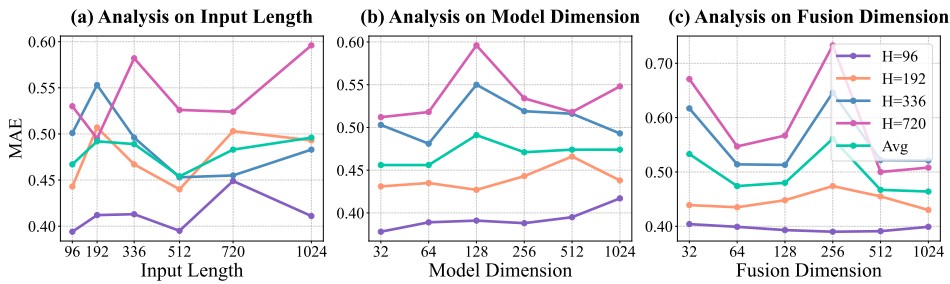

Figure 5: Hyperparameter sensitivity analysis on the ETTh1. The results illustrate the impact of input sequence length (left), model dimension (middle), and fusion dimension (right) on performance.

**Ablation Study.** Table 5 presents ablation studies on key components of LDM4TS. Both vision encoder and temporal encoder prove to be crucial, with their removal leading to significant performance degradation (18.00% and 18.42% MSE increase respectively), validating that our visual representations successfully capture essential temporal characteristics. The latent diffusion also plays a vital role (12.66% MSE increase when removed), demonstrating effective bridging between image reconstruction and time series prediction. Furthermore, removing individual transformation methods results in performance drops (5.78%, 5.47%, and 7.60% respectively), confirming that each view captures complementary temporal information to enhance forecasting performance. This validates our multi-view strategy that extracts and preserves diverse temporal features within the visual space.

**Parameter Sensitivity Analysis.** We further performed a parameter sensitivity analysis to investigate the effect of key hyperparameters on the model performance, as shown in Figure 5(a) shows the best performance at around 512 timesteps as input sequence length, while the performance of longer sequences decreases due to increased noise. The hidden dimension shows an optimum point between 32 and 64, balancing model capacity and risk of overfitting. For the hidden dimension values between 64 and 128 produce better results, suggesting that compact representations are more effective for integrating cross-modal information.

## 5 CONCLUSION

In this paper, we present LDM4TS, a novel framework that adapts the latent diffusion model with cross-modal conditional-guided mechanism for time series forecasting. By transforming temporal data into multi-view visual representations and reconstructing future images, LDM4TS effectively bridges the strengths of visual feature extraction and probabilistic generative modeling. Extensive experiments demonstrate that our method significantly outperforms existing diffusion-based methods and specialized forecasting models and excels at various forecasting tasks, providing a novel vision-enhanced perspective to address the key challenges of intrinsic temporal pattern extraction and uncertainty modeling. Future work will focus on exploring diffusion models' potential in broader time series applications and developing comprehensive benchmarks.

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

# A    EXPERIMENTAL DETAILS

## A.1    DATASET DETAILS

Table 6: Summary of the benchmark datasets. Each dataset contains multiple time series (Dim.) with different sequence lengths and is split into training, validation and testing sets. The data are collected at different frequencies across various domains. "Uncert.Var." means uncertainty variation.

| Dataset | Dim. | Series Length | Dataset Size | Frequency | Domain | Uncert.Var. |
|---------|------|---------------|--------------|-----------|--------|-------------|
| ETTm1 | 7 | {96, 192, 336, 720} | (34465, 11521, 11521) | 15 min | Temperature | 2.53 |
| ETTm2 | 7 | {96, 192, 336, 720} | (34465, 11521, 11521) | 15 min | Temperature | 1.27 |
| ETTh1 | 7 | {96, 192, 336, 720} | (8545, 2881, 2881) | 1 hour | Temperature | 2.50 |
| ETTh2 | 7 | {96, 192, 336, 720} | (8545, 2881, 2881) | 1 hour | Temperature | 1.29 |
| Electricity | 321 | {96, 192, 336, 720} | (18317, 2633, 5261) | 1 hour | Electricity | 3.94 |
| Weather | 21 | {96, 192, 336, 720} | (36792, 5271, 10540) | 10 min | Weather | - |

We conduct experiments on the above real-world datasets to evaluate the performance of our proposed model and follow the same data processing and train-validation-test set split protocol used in TimesNet benchmark Wu et al. (2022), ensuring a strict chronological order to prevent data leakage. Different datasets require specific adjustments to accommodate their unique characteristics:

**ETT Dataset Kim et al. (2022)**    The Electricity Transformer Temperature (ETT) dataset consists of both hourly (ETTh) and 15-minute (ETTm) frequency data, with 7 variables ($enc\_in = dec\_in = c\_out = 7$) measuring transformer temperatures and related factors. For ETTh data, we set periodicity to 24 with hourly frequency, while ETTm data uses a periodicity of 96 with 15-minute intervals. Standard normalization is applied to each feature independently, and the model maintains the same architectural configuration across both temporal resolutions.

**ECL Dataset Wu et al. (2021)**    The electricity consumption dataset contains 321 variables monitoring power usage patterns. We employ robust scaling techniques to handle outliers and implement sophisticated missing value imputation strategies. The model incorporates adaptive normalization layers to address the varying scales of electricity consumption across different regions and time periods. The daily periodicity is preserved through careful temporal encoding, while the high feature dimensionality is managed through efficient attention mechanisms.

**Weather Dataset Wu et al. (2021)**    This multivariate dataset encompasses 21 weather-related variables, each with distinct physical meanings and scale properties. Our approach implements feature-specific normalization to handle the diverse variable ranges while maintaining their physical relationships. The model captures both daily and seasonal patterns through enhanced temporal encoding, with special attention mechanisms designed to model the complex interactions between different weather variables. We maintain consistent prediction quality across all variables through carefully calibrated cross-attention mechanisms.

## A.2    OPTIMIZATION SETTINGS

### A.2.1    MODEL ARCHITECTURE PARAMETERS

The core architecture of our diffusion-based model consists of several key components, each with specific parameter settings. The autoencoder pathway is configured with an image size of $64 \times 64$ and a patch size of 16, providing an efficient latent representation while maintaining temporal information. The diffusion process uses 20 timesteps with carefully tuned noise scheduling ($\beta_{start} = 0.00085, \beta_{end} = 0.012$) to ensure stable training.

For the transformer backbone, we employ a configuration with $d\_model = 256$ and 8 attention heads, which empirically shows strong performance across different datasets. The encoder-decoder structure uses 2 encoder layers and 1 decoder layer, striking a balance between model capacity and computational efficiency.

Table 7: Default Model Architecture Parameters

| Parameter | Default Value | Description |
|---|---|---|
| *Visual Representation Parameters* | | |
| image_size | 64 | Size of generated image representation |
| patch_size | 16 | Size of patches for input processing |
| grayscale | True | Whether to use grayscale images |
| *Diffusion Process Parameters* | | |
| training_timesteps | 20 | Number of diffusion training steps |
| inference_timesteps | 20 | Number of inference steps |
| num_samples | 100 | Samples generated for the distribution |
| beta_start | 0.00085 | Initial value of noise schedule |
| beta_end | 0.012 | Final value of noise schedule |
| use_ddim | True | Whether to use DDIM sampler |
| unet_layers | 1 | Number of layers in UNet |
| *Model Architecture Parameters* | | |
| d_model | 256 | Dimension of model hidden states |
| d_ldm | 256 | Hidden dimension of LDM |
| d_fusion | 256 | Dimension of gated fusion module |
| e_layers | 2 | Number of encoder layers |
| d_layers | 1 | Number of decoder layers |
| *Training Configuration* | | |
| freeze_ldm | True | Whether to freeze LDM parameters |
| save_images | False | Whether to save generated images |
| output_type | full | Type of output for ablation study |

Table 8: Default Training Parameters

| Parameter | Default Value | Description |
|---|---|---|
| *Basic Training Parameters* | | |
| batch_size | 32 | Number of samples per training batch |
| learning_rate | 0.001 | Initial learning rate for optimization |
| train_epochs | 10 | Total number of training epochs |
| patience | 3 | Epochs before early stopping |
| loss | MSE | Type of loss function |
| label_len | 48 | Length of start token sequence |
| seq_len | 96/168(for probabilistic) | Length of input sequence |
| norm_const | 0.4 | Coefficient for normalization |
| padding | 8 | Size of sequence padding |
| stride | 8 | Step size for sliding window |
| pred_len | 96/192/336/720 | Available prediction horizons |
| *Dataset-specific Parameters* | | |
| c_out | 7 (ETTh1/h2/m1/m2) 21 (Weather) 321 (Electricity) 862 (Traffic) | Dataset-specific output dimensions |
| periodicity | 24 (ETTh1/h2/Electricity/Traffic) 96 (ETTm1/m2) 144 (Weather) | Natural cycle length per dataset |

### A.2.2 TRAINING PARAMETERS

We adopt a comprehensive training strategy with both general and task-specific parameters. The model is trained with a batch size of 32 and an initial learning rate of $0.001$, using the *AdamW* optimizer. Early stopping with a patience of 3 epochs is implemented to prevent over-fitting. For time series processing, we use a sequence length of 96 and a prediction length of 96, with a label length of 48 for teacher forcing during training.

The training process employs automatic mixed precision (AMP) when available to accelerate training while maintaining numerical stability. We use MSE as the primary loss function, supplemented by additional regularization terms for specific tasks.

### A.3 EVALUATION METRICS

For point forecasting evaluation metrics, we utilize the mean square error (MSE) and mean absolute error (MAE) to measure the accuracy of the predicted values compared to the ground truth. For probabilistic forecasting, we choose the quantile interval calibration error (QICE) to quantify the deviation between the proportion of true data contained within each interval and the optimal proportion. The calculations of these metrics are as follows:

$$\text{MSE} = \frac{1}{H} \sum_{h=1}^{H} (\mathbf{Y}_h - \hat{\mathbf{Y}}_h)^2, \quad \text{MAE} = \frac{1}{H} \sum_{h=1}^{H} |\mathbf{Y}_h - \hat{\mathbf{Y}}_h|, \quad \text{QICE} = \frac{1}{M} \sum_{m=1}^{M} |r_m - \frac{1}{M}|,$$

where $H$ denotes the number of data points (i.e., prediction horizon in our cases). $\mathbf{Y}_h$ and $\hat{\mathbf{Y}}_h$ are the $h$-th ground truth and prediction where $h \in \{1, \cdots, H\}$. For QICE, $r_m$ represents the actual coverage rate of the $m/M$-quantile interval, and $M$ is the number of quantile intervals evaluated (set to $M = 10$ in our experiments).

## B DETAILS OF BASELINE METHODS

We compare our approach with three categories of baseline methods used for comparative evaluation: transformer-based architectures, diffusion-based models, and other competitive approaches for time series forecasting.

**Transformer-based Models:** **FEDformer Zhou et al. (2022)** integrates wavelet decomposition with a Transformer architecture to efficiently capture multi-scale temporal dependencies by processing both time and frequency domains. **Autoformer Wu et al. (2021)** introduces a decomposing framework that separates the time series into trend and seasonal components, employing an autocorrelation mechanism for periodic pattern extraction. **ETSformer Woo et al. (2022)** extends the classical exponential smoothing method with a Transformer architecture, decomposing time series into level, trend, and seasonal components while learning their interactions through attention mechanisms. **Informer Zhou et al. (2021)** addresses the quadratic complexity issue of standard attention mechanisms through ProbSparse self-attention, which enables efficient handling of long input sequences. **Reformer Kitaev et al. (2020)** optimizes attention computation via Locality-Sensitive Hashing (LSH) and reversible residual networks, significantly reducing memory and computational costs. **PatchTST Nie et al. (2023b)** treats time series as a sequence of patches and employs a transformer architecture for long-term forecasting, showing strong performance through its patch-based approach. **Non-Stationary Transformer Liu et al. (2022b)** rethinks the stationarity assumption in time series forecasting by explicitly modeling non-stationary components within the Transformer framework. **TimeMixer++ Wang et al. (2024b)** enhances multiscale mixing capabilities through improved decomposition strategies and adaptive temporal fusion mechanisms.

**Diffusion-based Models:** **TimeGrad Rasul et al. (2021b)** pioneers diffusion for time series by incorporating autoregressive components for multivariate probabilistic forecasting. **CSDI Tashiro et al. (2021)** is tailored for irregularly-spaced time series, learning a score function of noise distribution under given conditions to generate samples for forecasting. **TimeDiff Shen & Kwok (2023)** introduces non-autoregressive conditional diffusion models for time series prediction, improving on previous autoregressive approaches. **TMDM Li et al. (2024)** employs transformer-modulated

diffusion models for probabilistic multivariate time series forecasting, combining the strengths of transformers and diffusion processes. **DiffusionTS Yuan & Qiao (2024)** presents an interpretable diffusion framework for general time series generation with enhanced controllability. **NsDiff Ye et al. (2025)** addresses non-stationary characteristics in time series through specialized diffusion modeling techniques. **ScoreGrad Song et al. (2020)** utilizes a continuous-time framework for progressive denoising from Gaussian noise to reconstruct the original signal, allowing for adjustable step sizes during the denoising process.

**Other Competitive Models:** **DLinear Zeng et al. (2023)** proposes a linear transformation approach directly on time series data, simplifying the prediction process under the assumption of linear changes over time. **TimesNet Wu et al. (2022)** focuses on multi-scale feature extraction using various convolution kernels to capture temporal dependencies of different lengths, automatically selecting the most suitable feature scales. **LightTS Campos et al. (2023)** aims to build lightweight time series forecasting models, streamlining structures and parameters to reduce computational resource requirements while maintaining high predictive performance. **iTransformer Liu et al. (2024)** is an inverted Transformer for TSF that embeds each variate's entire history as a variate token, and applies self-attention across variates to model multivariate correlations. **FITS Xu et al. (2023)** operates on the principle that time series can be manipulated through interpolation in the complex frequency domain. **VisionTS Chen et al. (2024)** leverages pre-trained vision models by transforming time series into visual representations. **Time-VLM Zhong et al. (2025)** explores multimodal vision-language models for time series forecasting by integrating temporal, visual, and textual modalities with frozen pre-trained VLMs.

Table 9: Probabilistic forecasting comparison by QICE (lower is better). Input length = 168, horizon = 192. We draw 100 samples per method to estimate predictive distributions.

| Dataset | TimeGrad (2021b) | CSDI (2021) | TimeDiff (2023) | TMDM (2024) | DiffusionTS (2024) | NsDiff (2025) | LDM4TS (ours) |
|---|---|---|---|---|---|---|---|
| ETTh1 | 6.731 | 3.107 | 14.931 | 2.821 | 6.423 | **1.470** | 1.589 |
| ETTh2 | 9.488 | 5.331 | 14.813 | 4.471 | 9.577 | 2.074 | **1.598** |
| ETTm1 | 6.693 | 2.828 | 14.795 | 2.567 | 5.605 | 2.041 | **1.590** |
| ETTm2 | 6.962 | 8.106 | 13.385 | 2.610 | 9.959 | 2.030 | **1.589** |
| ECL | 7.118 | 7.506 | 15.466 | 10.562 | 8.205 | 6.685 | **1.580** |

## C  PROBABILISTIC FORECASTING

While LDM4TS is primarily evaluated on point forecasting tasks, diffusion-based uncertainty modeling naturally extends to probabilistic forecasting. To evaluate the probabilistic forecasting capabilities of our model, we selected six strong baselines including TimeGrad Rasul et al. (2021a), CSDI Tashiro et al. (2021), TimeDiff Shen & Kwok (2023), TMDM Li et al. (2024), DiffusionTS Yuan & Qiao (2024) and NsDiff Ye et al. (2025). More details of these methods are in Appendix B. and employ the Quantile Interval Calibration Error (QICE) Han et al. (2022), which measures the calibration of prediction intervals across multiple quantiles. By leveraging the stochastic nature of the reverse diffusion process, we generate multiple samples that effectively quantify prediction uncertainty. As shown in Table 9, LDM4TS achieves superior performance on probabilistic metrics across most datasets. Especially on the ETT dataset family, LDM4TS consistently outperforms other previous diffusion-based methods, with a dramatic 10 times improvement compared to TimeDiff (1.589 vs 14.931) on ETTh1, and 43.77% improvement of CSDI on ETTm1 (1.590 vs 2.828). The advantages are particularly significant on larger datasets, where LDM4TS achieves only 23.63% QICE of the best baseline NsDiff (1.580 vs 6.685) on the ECL dataset.

### C.1  ANALYSIS OF PERFORMANCE ON IRREGULAR TIME SERIES

To further evaluate the robustness and generalizability of our approach, we conduct additional experiments on the Exchange rate dataset, which represents a particularly challenging scenario for time series forecasting due to its irregular temporal patterns and absence of clear periodic structures. Beyond the popular SOTA model PatchTST Nie et al. (2023b), we additionally compare recent

state-of-the-art models from 2024-2025, including iTransformer Liu et al. (2024), FITS Xu et al. (2023), and TimeMixer++ Wang et al. (2024b). We also compare against methods with pre-trained components, including VisionTS Chen et al. (2024) and Time-VLM Zhong et al. (2025), to ensure fair comparison within the same paradigm of leveraging pre-trained foundation models.

Table 10: Performance evaluation on irregular time series (Exchange dataset). The input sequence length is set to 96 for all baselines, and the average results of prediction lengths {96, 192, 336, 720} are reported.

| Dataset | LDM4TS (Ours) | | PatchTST (2023b) | | iTransformer (2024) | | FITS (2023) | | TimeMixer++ (2024b) | | VisionTS (2024) | | Time-VLM (2025) | |
|---------|------|------|------|------|------|------|------|------|------|------|------|------|------|------|
| Metric | MSE | MAE | MSE | MAE | MSE | MAE | MSE | MAE | MSE | MAE | MSE | MAE | MSE | MAE |
| *Exchange* | **0.393** | **0.416** | 0.427 | 0.436 | 0.441 | 0.457 | 0.458 | 0.457 | 0.471 | 0.467 | 0.483 | 0.461 | 0.555 | 0.481 |

The experimental results on the Exchange dataset demonstrate LDM4TS's superior performance on irregular time series, achieving MSE of 0.393 and MAE of 0.416, which represents 8.0% and 4.6% improvements over the second-best method PatchTST (0.427/0.436). Notably, our approach significantly outperforms other vision-enhanced methods, with 18.6% lower MSE than VisionTS (0.483) and 29.2% lower MSE than Time-VLM (0.555). These results validate that our multi-view transformation strategy and conditional diffusion framework effectively capture subtle temporal dependencies even in challenging datasets without clear periodic structures, extending the applicability of vision-enhanced forecasting to diverse real-world scenarios.

## D PREREQUISITES OF LATENT DIFFUSION MODELS

### D.1 AUTOENCODER FRAMEWORK

Latent Diffusion Models (LDMs) leverage the autoencoder architecture to facilitate efficient learning in the latent space. An autoencoder comprises two primary components: an encoder and a decoder. The encoder $\mathcal{E}$ compresses high-dimensional input data $x \in \mathbb{R}^D$ into a lower-dimensional latent representation $z \in \mathbb{R}^d$, where $d \ll D$. This compression not only reduces the computational complexity but also captures the essential features of the data. In our implementation, we utilize the pre-trained AutoencoderKL from the ***stable-diffusion-v1-4***, which has demonstrated remarkable capabilities in image compression and reconstruction. Mathematically, this process is described as:

$$z = \mathcal{E}(x) \tag{15}$$

**Latent Space Scaling** In practice, the latent representations produced by the encoder are typically scaled by a factor $s = 0.18215$ to ensure numerical stability and optimal distribution characteristics:

$$z_{scaled} = s \cdot \mathcal{E}(x) \tag{16}$$

This specific scaling factor originates from the VAE design in Stable Diffusion and is derived through empirical optimization. The value is calculated to minimize the KL divergence between the scaled latent distribution and the standard normal distribution:

$$s^* = \underset{s}{\arg\min} \, \mathbb{E}_{x \sim p_{data}}[D_{KL}(s \cdot \mathcal{E}(x) \| \mathcal{N}(0, 1))] \tag{17}$$

where $D_{KL}$ represents the Kullback-Leibler divergence. In our framework, this scaling operation serves multiple critical purposes. It ensures numerical stability during the diffusion process by maintaining consistent value ranges while facilitating better optimization dynamics by bringing the latent distribution closer to the standard normal. This operation also maintains compatibility with the pre-trained weights while allowing for efficient processing of our visual time series representations.

The optimization process involves collecting latent representations $z = \mathcal{E}(x)$ from a large dataset, computing their empirical statistics $\mu_z$ and $\sigma_z^2$, and determining the optimal scaling factor $s$ such that $s\sigma_z \approx 1$ to match the target standard deviation. This process has been extensively validated in the context of both image generation and, in our case, time series visual representations. During decoding, the inverse scaling is applied to restore the original magnitude:

$$\hat{x} = \mathcal{D}(z_{scaled}/s) \tag{18}$$

The autoencoder is trained to minimize the reconstruction loss:

$$\mathcal{L}_{\text{AE}} = \|\mathcal{D}(\mathcal{E}(x)) - x\|_2^2 \tag{19}$$

However, in the context of LDMs, the autoencoder enables operations to be performed in the compressed latent space, thereby enhancing efficiency without significant loss of information. In our implementation, we freeze the pre-trained autoencoder parameters in the LDM4TS model during training, focusing the optimization process on diffusion dynamics and temporal feature extraction. This design choice significantly reduces computational overhead while maintaining the benefits of well-learned representations from the compressed latent space.

## D.2 FOUNDATIONS OF DIFFUSION MODELS

Diffusion models define a principled framework for generative modeling through gradual noise addition and removal. In our LDM4TS framework, we adapt this process specifically for time series visual representations while maintaining the fundamental probabilistic structure.

**Forward Process**   The forward diffusion process follows a Markov chain that progressively adds Gaussian noise:

$$q(x_t|x_{t-1}) = \mathcal{N}(x_t; \sqrt{1 - \beta_t}x_{t-1}, \beta_t\mathbf{I}) \tag{20}$$

$$q(x_t|x_0) = \mathcal{N}(x_t; \sqrt{\bar{\alpha}_t}x_0, (1 - \bar{\alpha}_t)\mathbf{I}) \tag{21}$$

Here, $q(x_t|x_{t-1})$ describes the transition from step $t - 1$ to $t$, where $\beta_t$ controls the noise schedule. In our implementation, we adopt a linear noise schedule with carefully tuned parameters $\beta_{start} = 0.00085$ and $\beta_{end} = 0.012$. The second equation gives the direct relationship between any noisy sample $x_t$ and the original data $x_0$, where $\bar{\alpha}_t = \prod_{s=1}^{t}(1 - \beta_s)$ represents the cumulative product of noise levels.

**Reverse Process**   The reverse process learns to gradually denoise data through:

$$p_\theta(x_{t-1}|x_t) = \mathcal{N}(x_{t-1}; \mu_\theta(x_t, t), \Sigma_\theta(x_t, t)) \tag{22}$$

where the mean and variance are parameterized as:

$$\mu_\theta(x_t, t) = \frac{1}{\sqrt{\alpha_t}}(x_t - \frac{\beta_t}{\sqrt{1 - \bar{\alpha}_t}}\epsilon_\theta(x_t, t)) \tag{23}$$

$$\Sigma_\theta(x_t, t) = \frac{1 - \bar{\alpha}_{t-1}}{1 - \bar{\alpha}_t}\beta_t \tag{24}$$

In our framework, we modify the noise prediction network $\epsilon_\theta$ to accept additional conditioning information, transforming the reverse process into:

$$p_\theta(x_{t-1}|x_t, c) = \mathcal{N}(x_{t-1}; \mu_\theta(x_t, t, c), \Sigma_\theta(x_t, t)) \tag{25}$$

where $c$ represents the concatenated frequency domain embeddings and encoded textual descriptions. This modification allows the model to leverage both spectral and semantic information during the denoising process while maintaining the same variance schedule.

**Sampling Methods**   Different sampling strategies offer various trade-offs between generation quality and computational efficiency. In our implementation, we primarily utilize DDIM for its deterministic nature and faster sampling capabilities, though both approaches are supported:

- **DDPM**: Uses the full chain of $T$ steps with stochastic sampling:

$$x_{t-1} = \mu_\theta(x_t, t) + \sigma_t\epsilon, \quad \epsilon \sim \mathcal{N}(0, \mathbf{I}) \tag{26}$$

- **DDIM**: Enables faster sampling through deterministic trajectories:

$$x_{t-1} = \sqrt{\bar{\alpha}_{t-1}}\left(\frac{x_t - \sqrt{1 - \bar{\alpha}_t}\epsilon_\theta(x_t, t)}{\sqrt{\bar{\alpha}_t}}\right) + \sqrt{1 - \bar{\alpha}_{t-1}}\epsilon_\theta(x_t, t) \tag{27}$$

### D.3 U-NET ARCHITECTURE

The U-Net architecture serves as the backbone for noise prediction in our framework, combining multi-view processing with skip connections specifically designed for time series visual patterns. Our implementation modifies the standard U-Net structure to better handle temporal dependencies while maintaining spatial coherence.

**Encoder-Decoder Structure**    The architecture consists of multiple resolution levels:

- **Downsampling path**: Progressive feature compression

$$h_l = \text{ResBlock}(\text{Down}(h_{l-1})), \quad l = 1, \dots, L \tag{28}$$

- **Upsampling path**: Gradual feature reconstruction

$$h'_l = \text{ResBlock}(\text{Up}(h'_{l+1})) + h_l, \quad l = L, \dots, 1 \tag{29}$$

- **Skip connections**: Feature preservation across scales

$$h'_l = h'_l + \text{Project}(h_l) \tag{30}$$

**Feature Extraction**    Each resolution level processes features through a sequence of operations:

$$F_l = \text{Conv}(\text{GroupNorm}(\text{Attention}(h_l))) \tag{31}$$

These operations are augmented with timestep embeddings, which provide temporal information to guide the denoising process:

$$\gamma_t = \text{MLP}(\text{SinusoidalPos}(t)) \tag{32}$$

In our implementation, the timestep embedding is projected through a two-layer MLP with SiLU activation, following the design choices in Stable Diffusion for consistency and stability.

### D.4 CONDITIONAL GENERATION

Our framework implements a sophisticated dual-conditioning mechanism that leverages both frequency domain features and semantic descriptions to guide the diffusion process. This multi-modal approach enables robust capture of both temporal patterns and contextual information:

**Frequency Conditioning**    To effectively encode the rich spectral information inherent in time series data, we implement a sophisticated frequency domain transformation pipeline. This process begins with the application of a Hann window function, which is crucial for minimizing spectral leakage and enhancing frequency resolution:

$$w_t = 0.5(1 - \cos(\frac{2\pi t}{L-1})) \tag{33}$$

The frequency features are then systematically extracted through a carefully designed three-step process. First, we apply the window function to the input sequence:

$$X_{win} = X \odot w \tag{34}$$

Next, we transform the windowed signal into the frequency domain using the Fast Fourier Transform:

$$X_{fft} = \text{FFT}(X_{win}) = \sum_{t=0}^{L-1} X_{win}(t) \cdot e^{-2\pi ikt/L} \tag{35}$$

Finally, to preserve the complete spectral information, we concatenate the real and imaginary components of the FFT output:

$$c_{freq} = \text{Concat}[X_{fft_{real}}, X_{fft_{imag}}] \in \mathbb{R}^{B \times (2DL+2)} \tag{36}$$

where $L$ denotes the sequence length, $w$ represents the Hann window function, and $\odot$ indicates element-wise multiplication. The terms $X_{fft_{real}}$ and $X_{fft_{imag}}$ correspond to the real and imaginary components of the Fourier transform respectively. This comprehensive encoding strategy enables our model to capture both amplitude and phase information across multiple frequency bands, while maintaining computational efficiency through strategic dimensionality reduction.

**Text Conditioning**  To provide semantic guidance for the diffusion process, we automatically generate descriptive prompts by extracting key characteristics from the input time series. The prompt generation function $d_{prompt}(X)$ captures the following statistical properties:

- Statistical measures: minimum, maximum, and median values
- Temporal dynamics: overall trend direction and lag patterns
- Context information: prediction length and historical window size
- Domain knowledge: dataset-specific descriptions

These features are combined into a structured prompt template. A typical generated prompt follows the format:

> "$< |start\_prompt| >$Dataset description: {description}. Task: forecast the next {pred_len} steps given the previous {seq_len} steps. Input statistics: min value {min}, max value {max}, median value {median}, trend is {trend_direction}, top-5 lags are {lags}.$< |end\_prompt| >$"

The prompts are then processed through a frozen ***BERT-base-uncased*** model (110M parameters) to extract semantic features. Specifically, each prompt is first tokenized using BERT's WordPiece tokenizer with a maximum sequence length of 77 tokens:

$$h_{token} = \text{BERT}(d_{prompt}(X)) \in \mathbb{R}^{B \times L_{seq} \times d_{ff}} \tag{37}$$

where $L_{seq}$ is the sequence length after tokenization and $d_{ff} = 768$ is BERT's hidden dimension. The token-level features are aggregated through mean pooling to obtain a sequence-level representation:

$$h_{pool} = \frac{1}{L_{seq}} \sum_{i=1}^{L_{seq}} h_{token}[:, i, :] \in \mathbb{R}^{B \times d_{ff}} \tag{38}$$

The pooled features are then projected to match the latent dimension through a learnable transformation:

$$c_{text} = \text{TextEncoder}(X) = \text{TextProj}(h_{pool}) \in \mathbb{R}^{B \times d_{model}} \tag{39}$$

where $\text{TextProj}(\cdot)$ consists of a linear layer that projects from $d_{ff}$ to $d_{model}$, followed by layer normalization and ReLU activation to enhance feature expressiveness.

The frequency and text conditions are fused through a cross-modal attention mechanism:

$$c = \text{CrossAttn}(\text{MLP}([c_{\text{text}}; c_{\text{freq}}])) \in \mathbb{R}^{B \times d_{model}} \tag{40}$$

where the MLP first projects the concatenated features to a higher dimension for better feature interaction, and the cross-attention layer enables dynamic feature selection based on the latent representation. This combined conditioning signal guides the diffusion process by injecting both semantic and frequency information into each denoising step through the attention blocks of the U-Net architecture.

# E  ANALYSIS OF VISION TRANSFORMATION METHODS

Time series analysis faces the fundamental challenge of capturing complex temporal dynamics that manifest simultaneously across multiple scales. While traditional methods excel at specific temporal resolutions, they often struggle to comprehensively capture the full spectrum of patterns ranging from rapid local variations to gradual global trends. This limitation motivates our investigation into vision transformation techniques that can effectively encode rich temporal information into spatial patterns, making them amenable to powerful vision-based processing approaches.

Our framework introduces a systematic approach to time series visualization through theoretically-grounded transformation methods. Each method targets distinct yet complementary aspects of temporal dynamics, providing a comprehensive representation of the underlying time series structure. The transformation method we **implemented in the repository** is described below:

### E.1 SEGMENTATION REPRESENTATION (SEG)

The SEG transformation addresses the challenge of preserving local temporal structures while enabling efficient detection of periodic patterns. This method operates by restructuring a time series $x \in \mathbb{R}^L$ into a matrix $M \in \mathbb{R}^{\lceil L/T \rceil \times T}$, where $T$ represents the period length. The transformation process can be formally expressed as:

$$M_{i,j} = x_{(i-1)T+j}, \quad \text{for } i \in \{1, \ldots, \lceil L/T \rceil\}, j \in \{1, \ldots, T\}, \tag{41}$$

This segmentation approach offers several theoretical and practical advantages:

- **Local Structure Preservation:** Each row in the matrix represents a complete segment of length $T$, maintaining the original temporal relationships at the finest granularity.
- **Periodic Pattern Detection:** The vertical alignment of segments facilitates the identification of recurring patterns across different time periods.
- **Multi-scale Analysis:** By varying the period length $T$, the transformation can capture patterns at different temporal scales, enabling hierarchical pattern discovery.

The optimal period length T is determined through an optimization process that maximizes temporal correlation:

$$T = \arg \max_k \sum_{i=1}^{\lceil L/k \rceil} \sum_{j=1}^{k-1} \text{Corr}(M_{i,j}, M_{i,j+1}), \tag{42}$$

where $\text{Corr}(\cdot, \cdot)$ denotes the correlation between adjacent columns. This optimization ensures optimal alignment of periodic patterns while maintaining temporal fidelity.

### E.2 GRAMIAN ANGULAR FIELD (GAF)

The GAF transformation provides a sophisticated approach to encoding temporal relationships through polar coordinate mapping and trigonometric relationships. This method preserves both magnitude and temporal correlation information through a series of carefully designed transformations.

First, given a time series $\mathbf{x} = [x_1, x_2, \ldots, x_T] \in \mathbb{R}^T$, we normalize each element $x_i$ to a bounded interval. For the Gramian Angular Field, this normalization typically maps values to $[-1, 1]$ or $[0, 1]$ using min-max scaling:

$$\tilde{x}_i = \frac{x_i - \min(\mathbf{x})}{\max(\mathbf{x}) - \min(\mathbf{x}) + \epsilon}, \quad i \in 1, 2, \ldots, T, \tag{43}$$

where $\epsilon$ is a small constant (e.g., $10^{-8}$) added to prevent division by zero when the series has constant values. The normalized values $\tilde{x}_i$ are then encoded in a polar coordinate system. For each time step $i$, we compute:

$$\phi_i = \arccos(\tilde{x}_i), \quad r_i = \frac{i}{N}, \tag{44}$$

where $\phi_i$ represents the angular coordinate, $r_i$ represents the radial coordinate, and $N$ is a constant scaling factor that regularizes the span of the polar coordinates (typically $N = T$). The Gramian Angular Field (GAF) is then constructed as a matrix $\mathbf{G} \in \mathbb{R}^{T \times T}$ where each element $G_{i,j}$ encodes the trigonometric relation between points $(\phi_i, r_i)$ and $(\phi_j, r_j)$. For the Gramian Angular Summation Field (GASF) and Gramian Angular Difference Field (GADF), we have:

$$G_{i,j}^{\text{GASF}} = \cos(\phi_i + \phi_j) = \tilde{x}_i \tilde{x}_j - \sqrt{1 - \tilde{x}_i^2} \sqrt{1 - \tilde{x}_j^2}, \tag{45}$$

$$G_{i,j}^{\text{GADF}} = \sin(\phi_i - \phi_j) = \tilde{x}_j \sqrt{1 - \tilde{x}_i^2} - \tilde{x}_i \sqrt{1 - \tilde{x}_j^2}. \tag{46}$$

Both preserve the temporal correlation patterns in the original time series. The GAF transformation offers several key advantages:

- **Scale Invariance:** The polar encoding ensures that the representation is robust to amplitude variations.

- **Temporal Correlation Preservation:** The Gramian matrix captures both local and global temporal dependencies.

- **Dimensionality Reduction:** The transformation provides a compact representation while preserving essential temporal information.

### E.3    RECURRENCE PLOT (RP)

The RP transformation leverages phase space reconstruction to visualize the recurrent behavior in dynamical systems. Based on Taken's embedding theorem, this method first reconstructs the phase space trajectory:

$$\vec{x}_i = (x_i, x_{i+\tau}, ..., x_{i+(m-1)\tau}), \tag{47}$$

where $m$ is the embedding dimension and $\tau$ is the time delay. The recurrence matrix is then constructed as:

$$R_{i,j} = \Theta(\epsilon - \|\vec{x}_i - \vec{x}_j\|), \tag{48}$$

where $\Theta$ is the Heaviside function and $\epsilon$ is a threshold distance. This transformation reveals fundamental dynamical properties through several characteristic patterns:

- **Diagonal Lines:** Parallel to the main diagonal, indicating similar evolution of trajectories and revealing deterministic structures

- **Vertical/Horizontal Lines:** Representing periods of state stagnation or laminar phases

- **Complex Patterns:** Non-uniform structures indicating chaos or non-linear dynamics

### E.4    SPECTROGRAM - SHORT-TIME FOURIER TRANSFORM (STFT)

STFT provides a powerful representation of time series in the time-frequency domain, allowing for the analysis of how frequency content evolves over time. Unlike the standard Discrete Fourier Transform (DFT), which only describes the intensity $f(w)$ of each constituent frequency $w$ across the entire signal but lacks temporal localization, STFT addresses this limitation by computing localized frequency information within overlapping time windows. Given a time series $\mathbf{x} = [x_1, x_2, \ldots, x_T] \in \mathbb{R}^T$, the STFT is defined as:

$$\mathcal{F}(w, \tau) = \sum_{t=1}^{T} x_t g(t - \tau) e^{-iwt}, \quad S(w, \tau) = \log(1 + |\mathcal{F}(w, \tau)|^2), \tag{49}$$

where $w$ is the frequency variable, $\tau$ represents the position of the sliding window (time localization), $g(t)$ is a window function that confines the analysis to a local segment, and $\mathcal{F}(w, \tau)$ describes the complex amplitude of frequency $w$ at time step $\tau$. To generate a spectrogram image, we compute the power spectrum $|\mathcal{F}(w, \tau)|^2$ and often apply logarithmic scaling for vision modality normalization.

STFT spectrograms reveal periodic components and their temporal evolution, and separate noise into distinct frequency bands. The method highlights characteristic frequency signatures of temporal patterns and provides partial invariance to phase shifts and temporal warping.

### E.5    SPECTROGRAM - WAVELET

Wavelet Transform offers an alternative time-frequency representation that overcomes the fixed resolution limitations of STFT. By using basis functions (wavelets) that are localized in both time and frequency domains, this method provides multi-resolution analysis with adaptive time-frequency windows. The Continuous Wavelet Transform (CWT) of a time series $\mathbf{x} \in \mathbb{R}^T$ is defined as:

$$\mathcal{C}(s, \tau) = \int_{-\infty}^{\infty} x(t) \frac{1}{s} \psi^* \left( \frac{t - \tau}{s} \right) dt, \tag{50}$$

where $s$ is the scale parameter controlling frequency resolution, $\tau$ is the translation parameter indicating time position, $\psi^*$ is the complex conjugate of the mother wavelet, and $\mathcal{C}(s, \tau)$ represents the wavelet coefficient at scale $s$ and position $\tau$. In discrete implementation, this becomes:

$$\mathcal{C}(s, \tau) = \sum_{t=1}^{T} x_t \frac{1}{s} \psi^* \left( \frac{t - \tau}{s} \right). \tag{51}$$

The scale parameter $s$ has an inverse relationship with frequency—larger scales correspond to more stretched wavelets capturing lower frequencies, while smaller scales focus on higher frequencies and finer temporal details. The wavelet scalogram is created by computing $|\mathcal{C}(s,\tau)|^2$ and visualizing it as a heatmap with scales (or equivalent frequencies) on the vertical axis and time on the horizontal axis.

Wavelet transforms provide superior time-frequency localization compared to STFT, offer better resolution for transient events and rapid changes, adapt naturally to multi-scale patterns in data, and preserve important non-stationary characteristics that might be obscured in other transformations.

### E.6 Spectrogram - Mel Filterbank

Mel Filterbank transformation adapts spectral analysis to better align with human auditory perception, where frequency discrimination varies across the spectrum. Though originally designed for audio processing, this method offers valuable representations for general time series analysis.

Given a time series $\mathbf{x} \in \mathbb{R}^T$, the Mel Filterbank process begins with a pre-emphasis filtering step to amplify higher frequencies:

$$\hat{x}_t = x_t - \alpha x_{t-1}, \tag{52}$$

where $\alpha$ is a pre-emphasis coefficient (typically 0.95-0.97). Next, STFT is applied to obtain the power spectrum $|f(w,\tau)|^2$. The core innovation comes from applying a bank of $M$ triangular filters to the power spectrum, where these filters are spaced according to the Mel scale. The conversion from linear frequency $f$ to the Mel scale is given by:

$$m(f) = C_1 \log_{10}\left(1 + \frac{f}{C_2}\right), \tag{53}$$

where $C_1 = 2595$ and $C_2 = 700$ are Mel-scale constants derived from psychoacoustic research modeling human pitch perception. Each triangular filter $H_m(w)$ is centered at frequency $f_m$ on the Mel scale, with the filterbank output calculated as:

$$\hat{f}(m,\tau) = \sum_{w=0}^{N/2} |f(w,\tau)|^2 H_m(w), \quad \text{for } m = 1, 2, \ldots, M. \tag{54}$$

The resulting Mel spectrogram is visualized with Mel bands on the vertical axis and time on the horizontal axis, often with logarithmic compression:

$$S(m,\tau) = \log(\hat{f}(m,\tau) + \epsilon), \tag{55}$$

Mel Filterbank transformation captures perceptually relevant frequency information, reduces dimensionality while preserving essential spectral characteristics, emphasizes patterns in frequency ranges most critical for signal interpretation, and enhances detection of subtle spectral variations.

### E.7 Markov Transition Field (MTF)

The Markov Transition Field (MTF) transformation provides a principled approach to visualizing the dynamics of temporal transitions within a time series. By encoding the transition probabilities of quantized states, MTF captures both the temporal evolution and the underlying stochastic patterns of the sequence, making it particularly suitable for analyzing non-linear or non-stationary time series. Given a time series $\mathbf{x} = [x_1, x_2, \ldots, x_T]$, we first normalize and quantize the data:

$$q_t = Q\left(\frac{x_t - \min(\mathbf{x})}{\max(\mathbf{x}) - \min(\mathbf{x}) + \epsilon}\right), \quad t = 1, \ldots, T, \tag{56}$$

where $Q$ assigns each value to one of $n$ discrete bins, and $\epsilon$ is a small constant. The one-step Markov transition probability matrix $P \in \mathbb{R}^{n \times n}$ is estimated as:

$$P_{i,j} = \frac{\sum_{t=1}^{T-1} \mathbb{I}(q_t = i,, q_{t+1} = j)}{\sum_{t=1}^{T-1} \mathbb{I}(q_t = i)}, \tag{57}$$

where $\mathbb{I}(\cdot)$ is the indicator function. Finally, the Markov Transition Field $M \in \mathbb{R}^{T \times T}$ is constructed by mapping transition probabilities to all pairs of time steps:

$$M_{s,t} = P_{q_s, q_t}, \qquad 1 \leq s, t \leq T. \tag{58}$$

This process efficiently encodes the temporal dynamics of the quantized sequence into a two-dimensional field, suitable for visual analysis and downstream vision-based modeling. The MTF transformation exhibits several noteworthy advantages:

- **Stochastic Pattern Encoding:** By leveraging Markovian transition probabilities, MTF captures the probabilistic structure and dynamics of the underlying process.
- **Temporal Structure Preservation:** The two-dimensional field preserves temporal localization and global transition patterns simultaneously.
- **Compatibility with Vision Models:** The resulting matrix can be interpreted as an image, making it directly amenable to convolutional neural networks and other vision-based architectures.
- **Parameter Flexibility:** The resolution and sensitivity of the representation can be tuned by adjusting the number of quantization bins $n$, allowing adaptation to different data types and noise levels.

For long time series, computational efficiency can be improved via downsampling or segment-wise processing, with minimal loss of essential transition information. The MTF transformation thus offers a powerful and flexible mechanism for encoding both local and global temporal dependencies in a unified visual format, facilitating downstream tasks such as classification, anomaly detection, and similarity analysis.

## F    EFFICIENCY ANALYSIS OF DIFFUSION FORWARDS

Table 11: Computational efficiency on ETTh1. We report trainable parameters and per-batch inference latency (milliseconds) across prediction horizons $H$. Lower latency is better.

| Model | # Params | Inference Time (ms) | | | |
|---|---|---|---|---|---|
| | | $H=96$ | $H=192$ | $H=336$ | $H=720$ |
| **LDM4TS (Ours)** | **5.4M** | **76.88** | **80.31** | **193.44** | **192.19** |
| TimeGrad | 3.1M | 870.20 | 1854.50 | 3119.70 | 6724.10 |
| CSDI | 10.0M | 90.40 | 142.80 | 398.90 | 513.10 |
| SSSD | 32.0M | 418.60 | 645.40 | 1054.20 | 2516.90 |

We evaluate computational efficiency on ETTh1 by measuring per-batch inference latency across multiple horizons and by comparing model sizes with strong diffusion-based baselines. As summarized in Table 11, LDM4TS attains consistently low latency despite operating a generative diffusion backbone: (i) at short horizons ($H=96, 192$), LDM4TS is $11.3\times$–$23.1\times$ faster than TimeGrad and $5.5\times$–$8.0\times$ faster than SSSD; (ii) at long horizons ($H=336, 720$), it remains $16.1\times$–$35.0\times$ faster than TimeGrad and $5.5\times$–$13.1\times$ faster than SSSD. Compared to CSDI, LDM4TS is markedly faster at $H=336, 720$, while being competitive at short horizons.

## G    ANALYSIS OF TEXTUAL CONDITIONING

In this section, we conduct a detailed analysis of how textual conditioning influences the diffusion process and overall forecasting performance. While our primary experiments use ***BERT-based-uncased*** to encode statistical and domain descriptions as textual embedding, we investigate multiple variants to understand the optimal approach for integrating language representations into time series forecasting. *We aim to illustrate the full utilization of the textual modalities and flexibility of our proposed method.*

### G.1    EFFECT OF NUMERIC INFORMATION IN TEXTUAL PROMPTS

Textual conditioning (TC) can substantially improve forecasting, yet we observed occasional variability and **even degradations** when naively applying generic language embeddings. A plausible

cause is that off-the-shelf language models are not inherently optimized for encoding numeric tokens that are critical in TSF (e.g., range statistics, medians, nominal horizons). We therefore conduct a controlled study on ETTh1 to quantify the role of explicit numeric cues in prompts.

Concretely, we compare three variants on the ETTh1 dataset: (i) *w. TC*, where prompts include dataset- and instance-specific numeric descriptors (min/max/median, trend, horizon); (ii) *w/o. TC*, where textual conditioning is entirely removed; and (iii) *TC w/o number*, where prompts retain structure but strip numeric values. We also substitute different text encoders (BERT-base-uncased, GPT-2-small, RoBERTa-base) to assess encoder choice. As reported in Table 12, adding numeric information consistently improves both MSE and MAE over the non-numeric variant, confirming that numerical summaries provide salient contextual signals that generic text alone fails to capture. The gain becomes especially pronounced at long horizons (H=720), where numeric prompts reduce MSE by 29.4% relative to non-numeric prompts.

Table 12: ETTh1: Impact of numeric cues in prompts and choice of text encoder. Lower is better. Best and second-best per row are highlighted.

| Horizon | w. TC | | w/o. TC | | TC w/o number | | BERT-base | | GPT-2-small | | RoBERTa-base | |
|---|---|---|---|---|---|---|---|---|---|---|---|---|
| | MSE | MAE | MSE | MAE | MSE | MAE | MSE | MAE | MSE | MAE | MSE | MAE |
| 96 | **0.388** | **0.411** | 0.426 | 0.433 | 0.425 | 0.433 | 1.652 | 0.962 | 0.483 | 0.455 | 0.544 | 0.480 |
| 192 | **0.412** | **0.430** | 0.425 | 0.444 | 0.439 | 0.448 | 0.526 | 0.491 | 0.527 | 0.479 | 0.462 | 0.470 |
| 336 | 0.471 | 0.473 | 0.479 | 0.472 | **0.461** | **0.455** | 0.666 | 0.592 | 0.456 | 0.469 | 0.521 | 0.484 |
| 720 | **0.501** | **0.502** | 0.769 | 0.592 | 0.710 | 0.577 | 1.652 | 0.962 | 1.041 | 0.737 | 1.097 | 0.732 |
| Avg | **0.443** | **0.454** | 0.525 | 0.485 | 0.509 | 0.479 | 0.843 | 0.627 | 0.627 | 0.535 | 0.656 | 0.542 |

## G.2 COMPARISON OF DIFFERENT LANGUAGE MODELS

As shown in Table 12, we conducted experiments to compare how different pre-trained language models perform when their embeddings are used as the entire conditioning embedding input for time series forecasting. We experimented with three widely-used models: *BERT-base-uncased* (default), *GPT-2-small*, and *RoBERTa-base*. The results reveal that textual conditioning is crucial for our model, as removing it leads to significant performance degradation, particularly for long-horizon forecasting (H=720) where MSE increases from 0.501 to 0.769. Among the three language models compared, both *GPT-2-small* (average MSE=0.627) and *RoBERTa-base* (average MSE=0.656) outperform *BERT-base-uncased* (average MSE=0.843), with *BERT-base-uncased* showing particularly poor performance for long-term predictions (MSE=1.652). Despite these results, we chose BERT as our default encoder to minimize redundancy in textual information, as its simpler architecture is sufficient for extracting essential statistical patterns without overfitting linguistic nuances. However, our approach is flexible for easy substitution of text encoders, suggesting promising future directions for exploring specialized language models pre-trained on time series data. While integrating language models with time series forecasting remains challenging, properly implemented textual conditioning serves as a valuable complementary signal to frequency-domain features, particularly for complex time series with domain-specific characteristics.

# H STATEMENT ON LLM USAGE

We use LLMs solely to aid and polish writing, including covering spell checking, grammar fixes, style refinement, and minor wording suggestions. LLMs did not contribute to any scientific or technical content: all conceptualization, method design, implementation, experiments, result analysis, figures/tables, and conclusions were performed and verified by the authors. All cited works were independently retrieved, fully read, and manually verified using official sources; LLMs were never treated as authoritative references and were not used to generate or fabricate citations or results.

