# OpenReview forum: "Multi-view Latent Diffusion Reconstruction for Vision-enhanced Time Series Forecasting"
_ICLR.cc/2026/Conference — ICLR 2026 Conference Withdrawn Submission_

### Official Review · Reviewer_27Nf · 2025-10-25

**Soundness:** 1
**Presentation:** 2
**Contribution:** 2
**Rating:** 2
**Confidence:** 5

**Summary:**

This paper proposes LDM4TS, a vision-enhanced time series forecasting framework that transforms 1D temporal data into multi-view 2D visual representations, which are then reconstructed using a latent diffusion model. The reconstructed latent features are fused with temporal features for final predictions.

**Strengths:**

1. The paper introduces an interesting integration of multi-view transformations and latent diffusion-based reconstruction, bridging recent advances from vision and generative modeling domains.
2. The authors provide an insightful observation that 2D representations of time series can naturally reveal cross-periodic and structural correlations, which is indeed a meaningful perspective.

**Weaknesses:**

1. Experiment is unconvincing.
   - Although the paper claims extensive comparison, several key baselines are missing in the results table, such as recent vision-based or multimodal time-series models, such as VisionTS, TimeVLM, and DMMV, which are directly relevant to this line of research.
   - The benchmark data selection is limited, omitting more diverse datasets (e.g., Illness, Solar) that are commonly used to evaluate model generality.
   - The paper lacks forecast-horizon-specific results (e.g., per-horizon metrics like 96/192/336/720), making it hard to assess the robustness across varying prediction lengths.
2. Model design appears over-engineered and poorly justified.
   - The architecture combines latent diffusion, multiple-view visual transformations, and manually defined conditioning signals (frequency/text), yet the paper fails to clearly explain the design motivations and expected roles of each component.

**Questions:**

1. What is the true benefit of using latent diffusion reconstruction, how does it differ from using a masked autoencoding approach (e.g., VisionTS) for image-based reconstruction?
2. The authors claim that LDM4TS is the first to convert raw time series into multi-view visual representations, which is inaccurate — TimeVLM and other recent works already adopt multi-view with vision designs.
3. Diffusion-based reconstruction typically incurs high training and inference costs. Has the paper measured this overhead or compared runtime against simpler visual forecasting baselines?
4. Given the maturity of pre-trained 2D diffusion models, why not utilize existing pre-trained backbones or newer alternatives such as flow-matching or consistency models for efficient image reconstruction?

---

### Official Review · Reviewer_ncRQ · 2025-10-29

**Soundness:** 3
**Presentation:** 2
**Contribution:** 2
**Rating:** 4
**Confidence:** 4

**Summary:**

This paper proposes **LDM4TS**, a framework for time series forecasting that integrates a latent diffusion model with a cross-modal conditional guidance mechanism. The method first transforms time series data into multi-view visual representations (mainly using SEG, GAF, and RP transformations) to capture temporal dependencies across multiple scales. A latent diffusion model is then applied, conditioned on both frequency-domain and textual semantic information. Finally, a gated fusion mechanism integrates visual and temporal representations for forecasting. Experiments demonstrate that LDM4TS outperforms existing baselines across several standard benchmarks, especially in long-horizon, few-shot, and zero-shot forecasting scenarios.

**Strengths:**

- **Well-structured framework:** The model architecture is complete and logically coherent, consisting of three major components — multi-view encoding, latent diffusion-based generation, and multi-modal fusion.
- **Reasonable multi-modal design:** The use of multiple conditional sources (frequency-domain and semantic information) to guide the diffusion process is well-motivated and technically sound.
- **Comprehensive experiments:** The paper includes extensive evaluations across multiple datasets, ablation studies, parameter sensitivity analysis, and generalization tests, which together demonstrate the robustness and effectiveness of the proposed method.

**Weaknesses:**

- **Limited novelty:** The contribution appears to lie primarily in the engineering integration of existing techniques (visual encoding, diffusion modeling, and multimodal conditioning) rather than introducing fundamentally new modeling concepts.
- **Complex design but insufficient clarity:** Several core implementation details are under-explained. For example, the **Text Encoder** lacks a clear description of its semantic input and its role across datasets — does it directly take raw time series data as textual input, which would seem conceptually questionable? The **Temporal Projection** module is also insufficiently defined — what exact feature representation does it consume, and from which stage of the pipeline? Furthermore, it remains unclear whether the **Encoder/Decoder** bridging pixel and latent spaces are pre-trained or trained from scratch, and whether they are fine-tuned or kept frozen during training.
- **Lack of complexity analysis:** The paper does not quantify the computational cost or inference efficiency. Given the multi-stage pipeline that converts time series into images and employs diffusion sampling, a discussion on training and inference complexity compared to baseline models would be valuable.

**Questions:**

See Weaknesses. And an additional one:

Regarding the **Temporal Projection** branch: since it connects directly to the output through gated fusion, it appears to partially bypass the diffusion model.

- How does this affect the contribution of the diffusion component?
- What are the typical gating weights observed during inference?
- If the gate frequently favors one branch over the other, does that diminish the importance of the diffusion pathway?

An ablation study that disables the gate and retains only one of the two branches would help clarify the relative contribution of each module.

---

### Official Review · Reviewer_XQFv · 2025-10-30

**Soundness:** 2
**Presentation:** 2
**Contribution:** 2
**Rating:** 2
**Confidence:** 3

**Summary:**

This paper proposes an architecture for multimodal information extraction and its application in time series forecasting.

**Strengths:**

Combining multimodal methods to process time series is a direction worth exploring

**Weaknesses:**

1. The combination of multimodal information processing methods is certainly an interesting attempt, but the author did not provide a reasonable insight. Converting time series into images is a lossy transformation. Why is such a transformation beneficial? At the same time, the statistical characteristics of the data can be obtained through simple transformations, and adding textual information for description makes it more redundant.

2. Lack of a unique combination mechanism between time series and multimodal data, and the design of network structures mostly follows existing research.

3. The introduction of diffusion models is very incremental and marginal, and there are already many existing diffusion model methods for probabilistic forecasting. I am not sure about the specific significance of introducing diffusion models

4. The comparison objects are not up-to-date enough, and many new methods in time series and diffusion domain, such as [1] and [2], have not been considered.

[1] Liu Y, Hu T, Zhang H, et al. iTransformer: Inverted Transformers Are Effective for Time Series Forecasting[C]//The Twelfth International Conference on Learning Representations.

[2] Wang C, Yang L, Wang Z, et al. A Non-isotropic Time Series Diffusion Model with Moving Average Transitions[C]//Forty-second International Conference on Machine Learning.

**Questions:**

1. What is the significance of comparing zero-shot learning in Table 4? My understanding is that the author considered the zero-shot performance of the model because they used a pre-trained structure. However, the core component of the time series module is not pre-trained,  and this is actually more of a comparison of near shot transfer rather than zero shot. In addition, I am concerned that the zero-shot performance of the model comes from pre-trained models, which has nothing to do with whether the author's design can capture temporal information well. In other words, the zero-shot performance of the model is not the author's contribution.

2. If my understanding is correct, Table 12 mainly shows whether there are specific statistical features in the text that have an impact. If so, the statistical features of the time series should be very easy to obtain. Why is it necessary to introduce a text model with a huge number of parameters?

---

### Official Review · Reviewer_yQCM · 2025-11-01

**Soundness:** 2
**Presentation:** 3
**Contribution:** 3
**Rating:** 6
**Confidence:** 4

**Summary:**

The paper introduces LDM4TS, a forecasting framework that converts multivariate time series into multiple 2D visual representations and reconstructs them in a latent space using a conditional diffusion model.   The reconstructed visual features are fused with projected temporal features via a gated fusion module, and frequency-domain and text-like conditioning signals are used to inject global periodic and semantic priors and to support uncertainty estimation.   The model is evaluated on seven public benchmarks under full-data, few-shot, and zero-shot transfer, against Transformer-based, linear/decomposition, vision-enhanced, and diffusion-style baselines.  The paper reports lower or comparable MSE/MAE in most regimes and supports these claims with ablations and sensitivity analyses.

**Strengths:**

1. The paper presents an end-to-end pipeline unifying multiview time-series-to-vision encoding, latent diffusion with multimodal conditioning, and gated fusion for prediction, forming a single forecasting framework.
2. The model injects global periodic structure and semantic priors via frequency-domain and text-like conditioning and is intended to support both point forecasts and uncertainty estimation.
3. The evaluation spans seven public benchmarks and multiple regimes (long-horizon, few-shot, zero-shot transfer) and compares against Transformer-based, linear/decomposition, vision-enhanced, and diffusion-style baselines.
4. Reported results show lower or comparable MSE/MAE in many settings, including high-dimensional electricity load and cross-dataset transfer, and are backed by ablations and model-size sensitivity studies.

**Weaknesses:**

1. Although the paper claims calibrated predictive uncertainty through a generative diffusion mechanism, the main text only reports calibration metrics such as QICE for a single forecast horizon and does not provide a deeper analysis of uncertainty quality across different horizons, high-noise or distribution-shift regimes.
2. The zero-shot transfer results indicate maintained performance across datasets, but the study does not isolate how much generalization arises from the architectural design versus conditioning signals such as frequency-domain descriptors and text-like prompts provided at inference time, so the source of the observed transferability is not fully disentangled.
3. The paper reports performance (in MSE and MAE metrics) gains over multiple baselines and horizons but does not provide confidence intervals or significance tests for MSE and MAE differences, making it difficult to assess whether the average reported improvements are consistently reliable rather than specific to certain datasets or horizons.

**Questions:**

1. During inference, are the frequency-domain and textual/statistical conditioning signals strictly derived from past context only, or can they encode information that summarizes or implicitly reflects the target forecast interval, and if so how is information leakage prevented?
2. Can you provide an ablation in the zero-shot transfer setting where the model is evaluated without frequency or text conditioning, in order to separate architecture-driven generalization from prior injection via conditioning prompts?
3. The appendix reports QICE-based calibration results only for a single forecast horizon. Can you provide calibration and coverage metrics (e.g., QICE or interval coverage) across multiple forecast horizons to assess whether uncertainty quality is consistent as horizon increases?
4. The qualitative visualizations show predicted trajectories but do not include side-by-side comparisons with representative baselines. Can you add qualitative comparisons against strong baselines (e.g., a Transformer forecaster or a diffusion-style forecaster) to reveal systematic error differences?

---

### Note · Authors · 2025-11-19

I have read and agree with the venue's withdrawal policy on behalf of myself and my co-authors.